# From Posterior Sampling to Meaningful Diversity in Image Restoration

**Noa Cohen**
Technion – Israel Institute of Technology
noa.cohen@campus.technion.ac.il

**Hila Manor**
Technion – Israel Institute of Technology
hila.manor@campus.technion.ac.il

**Yuval Bahat**
Princeton University
yuval.bahat@gmail.com

**Tomer Michaeli**
Technion – Israel Institute of Technology
tomer.m@ee.technion.ac.il

## Abstract

Image restoration problems are typically ill-posed in the sense that each degraded image can be restored in infinitely many valid ways. To accommodate this, many works generate a diverse set of outputs by attempting to randomly sample from the posterior distribution of natural images given the degraded input. Here we argue that this strategy is commonly of limited practical value because of the heavy tail of the posterior distribution. Consider for example inpainting a missing region of the sky in an image. Since there is a high probability that the missing region contains no object but clouds, any set of samples from the posterior would be entirely dominated by (practically identical) completions of sky. However, arguably, presenting users with only one clear sky completion, along with several alternative solutions such as airships, birds, and balloons, would better outline the set of possibilities. In this paper, we initiate the study of **meaningfully diverse** image restoration. We explore several post-processing approaches that can be combined with any diverse image restoration method to yield semantically meaningful diversity. Moreover, we propose a practical approach for allowing diffusion based image restoration methods to generate meaningfully diverse outputs, while incurring only negligent computational overhead. We conduct extensive user studies to analyze the proposed techniques, and find the strategy of reducing similarity between outputs to be significantly favorable over posterior sampling. Code and examples are available on the project's webpage.

## 1 Introduction

Image restoration is a collective name for tasks in which a corrupted or low resolution image is restored into a better quality one. Example tasks include image inpainting, super-resolution, compression artifact reduction and denoising. Common to most image restoration problems is their ill-posed nature, which causes each degraded image to have infinitely many valid restoration solutions. Depending on the severity of the degradation, these solutions may differ significantly, and often correspond to diverse semantic meanings (Bahat & Michaeli, 2020).

In the past, image restoration methods were commonly designed to output a single solution for each degraded input (Haris et al., 2018; Kupyn et al., 2019; Liang et al., 2021; Pathak et al., 2016; Wang et al., 2018; Zhang et al., 2017). In recent years, however, a growing research effort is devoted to methods that can produce a range of different valid solutions for every degraded input, including in super-resolution (Bahat & Michaeli, 2020; Kawar et al., 2022; Lugmayr et al., 2022b), inpainting (Hong et al., 2019; Liu et al., 2021; Song et al., 2023), colorization (Saharia et al., 2022; Wang et al., 2022; Wu et al., 2021), and denoising (Kawar et al., 2021b; 2022; Ohayon et al., 2021). Broadly speaking, these methods strive to generate samples from the posterior distribution $P_{X|Y}$ of high-quality images $X$ given the degraded input image $Y$. Diverse restoration can then be achieved by repeatedly sampling from this posterior distribution. To allow this, significant research effort is devoted into approximating the posterior distribution, *e.g.*, using Generative Adversarial Networks

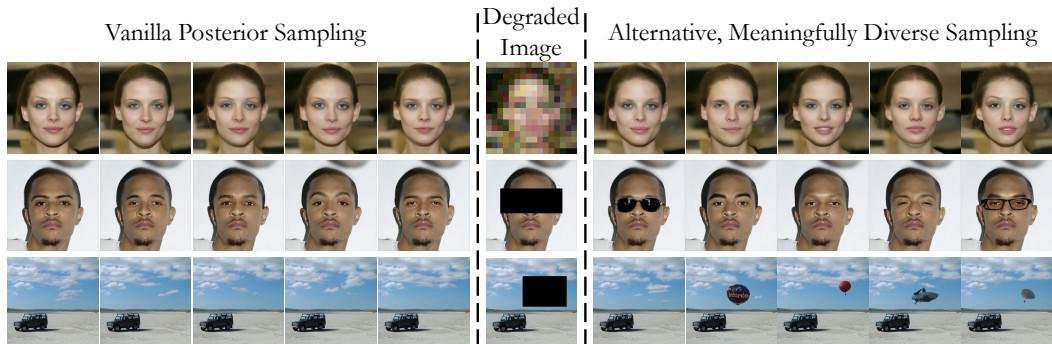

Figure 1: **Approximate posterior sampling vs. meaningfully diverse sampling in image restoration**. Restoration generative models aiming to sample from the posterior tend to generate images that highly resemble one another semantically (left). In contrast, the meaningful plausible solutions on the right convey a broader range of restoration possibilities. Such sets of restorations are achieved using the FPS approach explored in Sec. 4.

(GANs) (Hong et al., 2019; Ohayon et al., 2021), auto-regressive models (Li et al., 2022; Wan et al., 2021), invertible models (Lugmayr et al., 2020), energy-based models (Kawar et al., 2021a; Nijkamp et al., 2019), or more recently, denoising diffusion models (Kawar et al., 2022; Wang et al., 2022).

In this work, we question whether sampling from the posterior distribution is the optimal strategy for achieving *meaningful* solution diversity in image restoration. Consider, for example, the task of inpainting a patch in the sky like the one depicted in the third row of Fig. 1. In this case, the posterior distribution would be entirely dominated by patches of partly cloudy sky. Repeatedly sampling patches from this distribution would, with very high probability, yield interchangeable results that appear as reproductions for any human observer. Locating a notably different result would therefore involve an exhausting, oftentimes prohibitively long, re-sampling sequence. In contrast, we argue that presenting a set of alternative completions depicting an airship, balloons, or even a parachutist would better convey the actual possible diversity to a user.

Here we initiate the study of *meaningfully diverse image restoration*, which aims at reflecting to a user the perceptual range of plausible solutions rather than adhering to their likelihood. We start by analyzing the nature of the posterior distribution, as estimated by existing diverse image restoration models, in the tasks of inpainting and super-resolution. We show both qualitatively and quantitatively that this posterior is quite often heavy tailed. As we illustrate, this implies that if the number of images presented to a user is restricted to *e.g.*, 5, then with very high probability this set is not going to be representative. Namely, it will typically exhibit low diversity and will not span the range of possible semantics. We then move on to explore several baseline techniques for sub-sampling a large set of solutions produced by a posterior sampler, so as to present users with a small diverse set of plausible restorations. Finally, we propose a practical approach that can endow diffusion based image restoration models with the ability to produce small diverse sets. The findings of our analysis (both qualitatively and via a user study) suggest that techniques that explicitly seek to maximize distances between the presented images, whether by modifying the generation process or in post-processing, are significantly advantageous over random sampling from the posterior.

## 2 RELATED WORK

**Approximate posterior sampling.** Recent years have seen a shift from the one-to-one restoration paradigm to diverse image restoration. Methods that generate diverse solutions are based on various approaches, including VAEs (Peng et al., 2021; Prakash et al., 2021; Zheng et al., 2019), GANs (Cai & Wei, 2020; Liu et al., 2021; Zhao et al., 2020; 2021), normalizing flows (Helminger et al., 2021; Lugmayr et al., 2020) and diffusion models (Kawar et al., 2022; Lugmayr et al., 2022a; Wang et al., 2022). Common to these methods is that they aim for sampling from the posterior distribution of natural images given the degraded input. While this generates some diversity, in many cases the vast majority of samples produced this way have the same semantic meanings.

**Enhancing perceptual coverage.** Several works increase sample diversity in unconditional generation, *e.g.*, by pushing towards higher coverage of low density regions (Sehwag et al., 2022; Yu et al., 2020). For conditional generation, previous works attempted to battle the effect of the heavy-tailed nature of visual data (Sehwag et al., 2022) by encouraging exploration of the sample space during training (Mao et al., 2019). As we show, the approximated posterior of restoration models exhibits a similar heavy-tailed nature. For linear inverse problems, diversity can be increased, *e.g.*, by using geometric-based methods to traverse the latent space (Montanaro et al., 2022). However, these works do not improve on the redundancy when simultaneously sampling a batch from the heavy-tailed distribution (see *e.g.*, Fig. 1(d) in Sehwag et al. (2022), which depicts two pairs of very similar images within a set of 12 unconditional image samples). Our work is the first to explore ways to produce a *representative set* of meaningfully diverse solutions.

**Interactive exploration of solutions.** Another approach for conveying the range of plausible restorations is to hand over the reins to the user, by developing controllable methods. These methods allow the user to explore the space of possible restoration by various means, including graphical user interface tools (Bahat & Michaeli, 2020; 2021; Weber et al., 2020), editing of semantic maps (Buhler et al., 2020), manipulation in some latent space (Lugmayr et al., 2020; Wang et al., 2019), and via textual prompts describing a desired output (Bai et al., 2023; Chen et al., 2018; Ma et al., 2022; Zhang et al., 2020). These approaches are mainly suitable for editing applications, where the user has some end-goal in mind, and are also time consuming and require skill to obtain a desired result.

**Uncertainty quantification.** Rather than generating a diverse set of solutions, several methods present to the user a single prediction along with some visualization of the uncertainty around that prediction. These visualizations include heatmaps depicting per-pixel confidence-levels (Angelopoulos et al., 2022; Lee & Chung, 2019), as well as upper and lower bounds (Horwitz & Hoshen, 2022; Sankaranarayanan et al., 2022) that span the set of possibilities with high probability, either in pixel space or semantically along latent directions. However, per-pixel maps tend to convey little information about semantics, and latent space analyses require a generative model in which all attributes of interest are perfectly disentangled (a property rarely satisfied in practice).

## 3 LIMITATIONS OF POSTERIOR SAMPLING

When sampling multiple times from diverse restoration models, the samples tend to repeat themselves, exhibiting only minor semantic variability. This is illustrated in Fig. 2, which depicts two masked images with corresponding 10 random samples each, obtained from RePaint (Lugmayr et al., 2022a), a diverse inpainting method. As can be seen, none of the 10 completions corresponding to the eye region depict glasses, and none of the 10 samples corresponding to the mouth region depict a closed mouth. Yet, when examining 100 samples from the model, it is evident that such completions are possible; they are simply rare (2 out of 100 samples). This behavior is also seen in Figs. 1 and 5.

We argue that this phenomenon stems from the fact that the posterior distribution is often heavy-tailed along semantically interesting directions. Heavy-tailed distributions assign a non-negligible probability to distinct "outliers". In the context of image restoration, these outliers often correspond to different semantic meanings. This effect can be seen on the right pane of Fig. 2, which depicts the histogram of the projections of the 100 posterior samples onto their first principal component.

A quantitative measure of the tailedness of a distribution $P_X$ with mean $\mu$ and variance $\sigma^2$, is its kurtosis, $\mathbb{E}_{X \sim P_X}[((X - \mu)/\sigma)^4]$. The normal distribution family has a kurtosis of 3, and distributions with kurtosis larger than 3 are heavy tailed. As can be seen, both posterior distributions in Fig. 2 have very high kurtosis values. As we show in Fig. 3, cases in which the posterior is heavy tailed are not rare. For roughly 12% of the inspected masked face images, the estimated kurtosis value of the restorations obtained with RePaint was greater than 5, while only about 0.12% of Gaussian distributions over a space with the same dimension are likely to reach this value.

## 4 WHAT MAKES A SET OF RECONSTRUCTIONS MEANINGFULLY DIVERSE?

Given an input image $y$ that is a degraded version of some high-quality image $x$, our goal is to compose a set of $N$ outputs $\mathcal{X} = \{x^1, \cdots, x^N\}$ such that each $x^i$ constitutes a plausible reconstruction of $x$, while $\mathcal{X}$ as a whole reflects the diversity of possible reconstructions in a meaningful manner. By 'meaningful' we mean that rather than adhering to the posterior distribution of $x$ given $y$, we

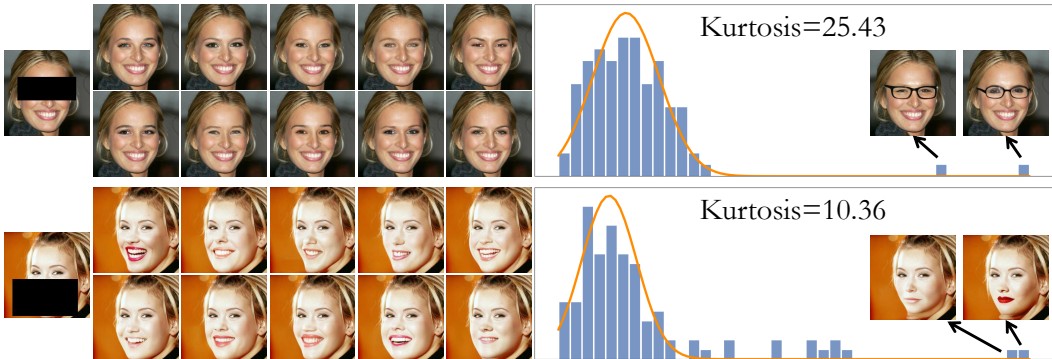

Figure 2: **Histograms of the projections of features from two collections of posterior samples onto their first principal component**. Each collection contains 100 reconstructions of an inpainted image. In the upper example PCA was applied on pixel space, and in the lower example on deep features of an attribute predictor. The high distribution kurtosis marked on the graphs are due to rare, yet non negligible points distant from the mean. We fit a mixture of 2 Gaussians to each distribution and plot the dominant Gaussian, to allow visual comparison of the tail.

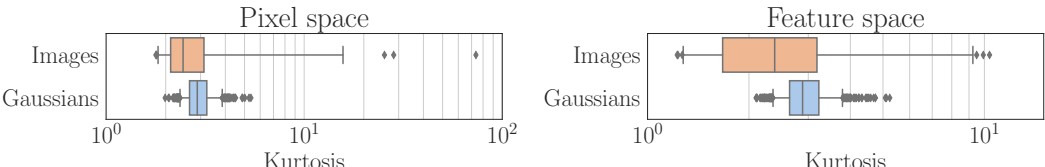

Figure 3: **Statistics of kurtoses in posterior distributions**. We calculate kurtoses values for projections of features from 54 collections of posterior samples, 100 samples each, onto their first principal component (orange). Left pane shows restored pixel values as features, while the right pane shows feature activations extracted from an attribute predictor. For comparison, we also show statistics of kurtoses of 800 multivariate Gaussians with the same dimensions (blue), each estimated from 100 samples. The non-negligible occurrence of very high kurtoses in images (compared with their Gaussian equivalents) indicates their heavy tailed distributions. Whiskers mark the $[5, 95]$ percentiles.

want $\mathcal{X}$ to *cover the perceptual range* of plausible reconstructions of $x$, to the maximal extent possible (depending on $N$). In practical applications, we would want $N$ to be small (*e.g.*, 5) to avoid the need of tedious scrolling through many restorations. Our goal in this section is to examine what mathematically characterizes a meaningfully diverse set of solutions. We do not attempt to devise a practical method yet, a task which we defer to Sec. 5, but rather only to understand the principles that should guide one in the pursuit of such a method. To do so, we explore three approaches for choosing the samples to include in the representative set $\mathcal{X}$ from a larger set of solutions $\tilde{\mathcal{X}} = \{\tilde{x}^1, \cdots, \tilde{x}^{\tilde{N}}\}$, $\tilde{N} \gg N$, generated by some diverse image restoration method. We illustrate the approaches qualitatively and measure their effectiveness in user studies. We note that a set of samples can either be presented to a user all at once or via a hierarchical structure (see App. G).

Given a degraded input image $y$, we start by generating a large set of solutions $\tilde{\mathcal{X}}$ using a diverse image restoration method. We then extract perceptually meaningful features for all $\tilde{N}$ images in $\tilde{\mathcal{X}}$ and use the distances between these features as proxy to the perceptual dissimilarity between the images. In each of the three approaches we consider, we use these distances in a different way in order to sub-sample $\tilde{\mathcal{X}}$ into $\mathcal{X}$, exploring different concepts of diversity. As a running example, we illustrate the approaches on the 2D distribution shown in Fig. 4, and on inpainting and super-resolution, as shown in Fig. 5 (see details in Sec. 4.1). Note that a small random sample from the distribution of Fig. 4 (second pane) is likely to include only points from the dominant mode, and thus does not convey to a viewer the existence of other modes. We consider the following approaches.

**Cluster representatives** A straightforward way to represent the different semantic modes in $\tilde{\mathcal{X}}$ is via clustering. Specifically, we apply the $K$-*means* algorithm over the feature representations of all images in $\tilde{\mathcal{X}}$, setting the number of clusters $K$ to the desired number of solutions, $N$. We then

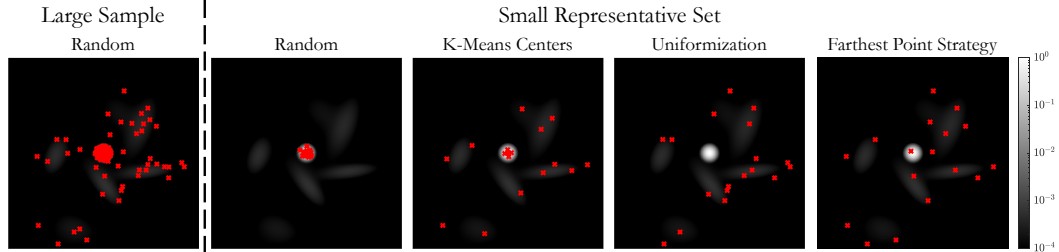

Figure 4: **Methods for choosing a small representative set**. We compare three baseline approaches for meaningfully representing a set $\tilde{\mathcal{X}}$ of $\tilde{N} = 1000$ red points drawn from an imbalanced mixture of 10 Gaussians (left), by using a subset $\mathcal{X}$ of only $N = 20$ points. Note how the presented approaches differ in their abilities to cover sparse and dense regions of the original set $\tilde{\mathcal{X}}$. In this example, $\tilde{\mathcal{X}}$ is dominated by the central Gaussian which contains 95% of the probability mass.

construct $\mathcal{X}$ by choosing for each of the $N$ clusters the image in $\tilde{\mathcal{X}}$ closest to its center in feature space. As seen in Figs. 4 and 5, this approach leads to a more diverse set than random sampling. However, the set can be redundant, as multiple points may originate from the dominant mode.

**Uniform coverage of the posterior's effective support**   In theory, one could go about our goal of covering the perceptual range of plausible reconstructions by sampling uniformly from the effective support of the posterior distribution $P_{X|Y}$ over a semantic feature space. This technique boils down to increasing the relative probability of sampling less likely solutions at the expense of decreasing the chances of repeatedly sampling the most likely ones, and we therefore refer to it as ***Uniformization***. This can be done by assigning to each member of $\tilde{\mathcal{X}}$ a probability mass that is inversely proportional to the density of the posterior at that point, and populating $\mathcal{X}$ by sampling from $\tilde{\mathcal{X}}$ without repetition according to these probabilities. Please refer to App. A for a detailed description of this approach. As seen in Figs. 4 and 5, an inherent limitation of this approach is that it may under-represent high-probability modes if their effective support is small. For example in Fig. 4, although Uniformization leads to a diverse set, this set does not contain a single representative from the dominant mode.

**Distant representatives**   The third approach we explore aims to sample a set of images that are as far as possible from one another in feature space, and relies on the ***Farthest Point Strategy (FPS)***, originally proposed for progressive image sampling (Eldar et al., 1997). The first image in this approach is sampled randomly from $\tilde{\mathcal{X}}$. With high probability, we can expect it to come from a dense area in feature space and thus to represents the most prevalent semantics in the set $\tilde{\mathcal{X}}$. The remaining $N-1$ images are then added in an iterative manner, each time choosing the image in $\tilde{\mathcal{X}}$ that is farthest away from the set constructed thus far. Note that here we do not aim to obtain a uniform coverage, but rather to sample a subset that maximizes the pairwise distances in some semantically meaningful feature space. This approach thus explicitly pushes towards semantic variability. Contrary to the previous approaches, the distribution of the samples obtained from FPS highly depends on the size of the set from which we sample. The larger $\tilde{N}$ is, the greater the probability that the set $\tilde{\mathcal{X}}$ contains extremely rare solutions. In FPS, these very rare solutions are likely to be chosen first. To control the probability of choosing improbable samples, FPS can be applied to a random subset of $L \leq \tilde{N}$ images from $\tilde{\mathcal{X}}$. As can be seen in Figs. 4 and 5, FPS chooses a diverse set of samples that on one hand covers all modes of the distribution (contrary to Uniformization) and on the other hand is not redundant (in contrast to $K$-means). Here we used $L = \tilde{N}$ (please see the effect of $L$ in App. B).

## 4.1   QUALITATIVE ASSESSMENT AND USER STUDIES

To assess the ability of each of the approaches discussed above to achieve meaningful diversity, we perform a qualitative evaluation and conduct a comprehensive user study. We experiment with two image restoration tasks: inpainting and noisy $16\times$ super-resolution with a bicubic down-sampling kernel and a noise level of 0.05. We analyze them in two domains: face images from the CelebAMask-HQ dataset (Lee et al., 2020) and natural images from the PartImagenet dataset (He et al., 2022). We use RePaint (Lugmayr et al., 2022a) and DDRM (Kawar et al., 2022) as our base diverse restoration models for inpainting and super-resolution, respectively. For faces, we use deep features of the AnyCost attribute predictor (Lin et al., 2021), which was trained to identify a range of facial features such as smile, hair color and use of lipstick, as well as accessories such as glasses.

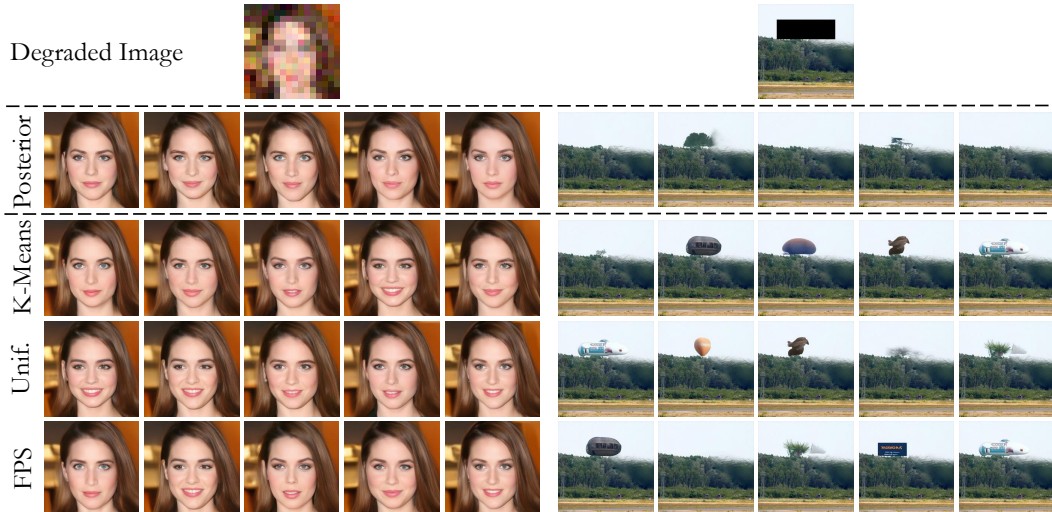

Figure 5: **Diversely sampling image restorations**. Using five images to represent sets of 100 restorations corresponding to degraded images (shown above), on images from CelebAMask-HQ (left) and PartImagenet (Right). The posterior subset (first row) is comprised of randomly drawn restoration solutions, while subsequent rows are constructed using the explored baselines.

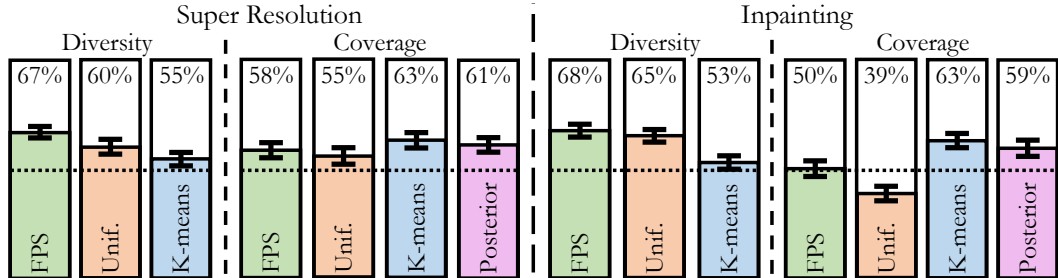

Figure 6: **Human perceived diversity and coverage of likely solutions.** For each domain, we report the percentage of users perceiving higher *diversity* in the explored sampling approaches compared to sampling from the approximate posterior, and the percentage of users perceiving sufficient *coverage* by any of the sampling approaches (including vanilla sampling from the posterior). We use bootstrapping for calculating confidence intervals.

We reduce the dimensions of those features to 25 using PCA, and use $L^2$ as the distance metric. For PartImagenet, we use deep features from VGG-16 (Simonyan & Zisserman, 2015) directly and via the LPIPS metric (Zhang et al., 2018). For face inpainting we define four varied possible masks, and for PartImagenet we construct masks using PartImagenet segments. In all experiments, we use an initial set $\tilde{\mathcal{X}}$ of $\tilde{N} = 100$ images generated from the model, and compose a set $\mathcal{X}$ of $N = 5$ representatives (see App. C for more details).

Figures 1 and 5 (as well as the additional figures in App. I) show several qualitative results. In all those cases, the semantic diversity in a random set of 5 images sampled from the posterior is very low. This is while the FPS and Uniformization approaches manage to compose more meaningfully diverse sets that better cover the range of possible solutions, *e.g.*, by inpainting different objects or portraying diverse face expressions (Fig. 5). These approaches automatically pick such restorations, which exist among the 100 samples in $\tilde{\mathcal{X}}$, despite being rare.

We conducted user studies through Amazon Mechanical Turk (AMT) on both the inpainting and the super-resolution tasks, using 50 randomly selected face images from the CelebAMask-HQ dataset per task. AMT users were asked to answer a sequence of 50 questions, after completing a short tutorial comprising two practice questions with feedback. To evaluate whether subset $\mathcal{X}$ constitutes a meaningfully diverse representation of the possible restorations for a degraded image, our study

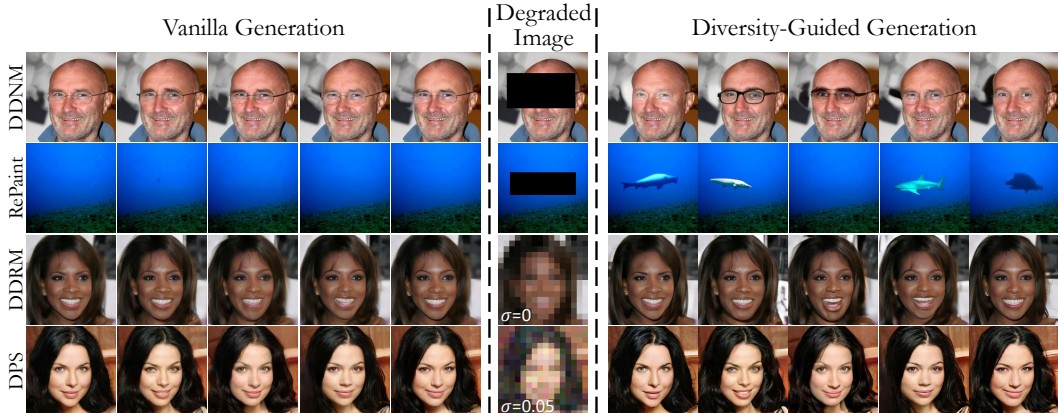

Figure 7: **Generating diverse image restorations.** Qualitative comparison of 5 restorations corresponding to degraded images (center) generated by the models specified on the left, without (left) and with (right) diversity guidance.

comprised two types of tests (for both restoration tasks). The first is a *paired diversity test*, in which users were shown a set of five images sampled randomly from the approximate posterior against five images sampled using one of the explored approaches, and were asked to pick the more diverse set. The second is an *unpaired coverage test*, in which we generated an additional (101[th]) solution to be used as a target image, and showed users a set of five images sampled using one of the four approaches. The users had to answer whether it includes at least one image very-similar to the target.

The results for both tests are reported in Fig. 6. As can be seen on the left pane, the diversity of approximate posterior sampling was preferred significantly less times than the diversity of any of the other proposed approaches. Among the three studied approaches, FPS was considered the most diverse. The results on the right pane suggest that all approaches, with the exception of Uniformization in inpainting, yield similar coverage for likely solutions, with a perceived similar image in approximately $60\%$ of the times. This means that the ability of the other two approaches (especially that of FPS) to yield meaningful diversity does not come at the expense of covering the likely solutions, compared with approximate posterior sampling (which by definition tends to present the more likely restoration solutions). In contrast, coverage by the Uniformization approach is found to be low, which aligns with the qualitative observation from Fig. 4.

Overall, the results from the two human perception tests confirm that, for the purpose of composing a meaningfully diverse subset of restorations, the FPS approach has a clear advantage over the $K$-means and Uniformization alternatives, and an even clearer advantage over randomly sampling from the posterior. While introducing a small drop in covering of the peak of the heavy-tailed distribution, it shows a significant advantage in terms of presenting additional semantically diverse plausible restorations. Please refer to App. E for more details on the user studies.

## 5  A PRACTICAL METHOD FOR GENERATING MEANINGFUL DIVERSITY

Equipped with the insights from Sec. 4, we now turn to propose a practical method for generating a set of meaningfully diverse image restorations. We focus on restoration techniques that are based on diffusion models, as they achieve state-of-the-art results. Diffusion models generate samples by attempting to reverse a diffusion process defined over timesteps $t \in \{0, \dots, T\}$. Specifically, their sampling process starts at timestep $t = T$ and gradually progresses until reaching $t = 0$, in which the final sample is obtained. Since these models involve a long iterative process, using them to sample $\tilde{N}$ reconstructions in order to eventually keep only $N \ll \tilde{N}$ images, is commonly impractical. However, we saw that a good sampling strategy is one that strives to reduce similarity between samples. In diffusion models, such an effect can be achieved using guidance mechanisms.

Specifically, we run the diffusion process to simultaneously generate $N$ images, all conditioned on the same input $y$ but driven by different noise samples. In each timestep $t$ within the generation process, diffusion models produce an estimate of the clean image. Let $\mathcal{X}_t = \{\hat{x}_{0|t}^1, \dots, \hat{x}_{0|t}^N\}$ be the

| Diversity | | Quality | |
|---|---|---|---|
| Super Resolution | 70% | Super Resolution | 63% |
| Inpainting | 60% | Inpainting | 46% |

Figure 8: **Human perceived diversity and quality.** We report on the left and right panes the percentage of users perceiving higher diversity and quality, respectively, in our diversity-guided generation process compared to the vanilla process. We calculate confidence intervals using bootstrapping.

set of $N$ predictions of clean images at time step $t$, one for each image in the batch. We aim for $\mathcal{X}_0$ to be the final restorations presented to the user (equivalent to $\mathcal{X}$ of Sec. 4), and therefore aim to reduce the similarities between the images in $\mathcal{X}_t$ at every timestep $t$. To achieve this, we follow the approach of Dhariwal & Nichol (2021), and add to each clean image prediction $\hat{x}_{0|t}^i \in \mathcal{X}_t$ the gradient of a loss function that captures the dissimilarity between $\hat{x}_{0|t}^i$ and its nearest neighbor within the set, $\hat{x}_{0|t}^{i,\text{NN}} = \arg\min_{x \in \mathcal{X}_t \setminus \{\hat{x}_{0|t}^i\}} d(\hat{x}_{0|t}^i, x)$, where $d(\cdot, \cdot)$ is a dissimilarity measure. In particular, we modify each prediction as

$$\hat{x}_{0|t}^i \leftarrow \hat{x}_{0|t}^i + \eta \frac{t}{T} \nabla d\left(\hat{x}_{0|t}^i, \hat{x}_{0|t}^{i,\text{NN}}\right), \qquad (1)$$

where $\eta$ is a step size that controls the guidance strength and the gradient is with respect to the first argument of $d(\cdot, \cdot)$. The factor $t/T$ reduces the guidance strength throughout the diffusion process.

In practice, we found the squared $L^2$ distance in pixel space to work quite well as a dissimilarity measure. However, to avoid pushing samples away from each other when they are already far apart, we clamp the distance to some upper bound $D$. Specifically, let $S$ be the number of unknowns in our inverse problem (*i.e.*, the number of elements in the image $x$ for super-resolution and the number of masked elements in inpainting). Then, we take our dissimilarity metric to be $d(u, v) = \frac{1}{2}\|u - v\|^2$ if $\|u - v\| \leq SD$ and $d(u, v) = \frac{1}{2}S^2 D^2$ if $\|u - v\| > SD$. The parameter $D$ controls the minimal distance from which we do not apply a guidance step (*i.e.*, the distance from which the predictions are considered dissimilar enough). Substituting this distance metric into (1), leads to our update step

$$\hat{x}_{0|t}^i \leftarrow \hat{x}_{0|t}^i + \eta \frac{t}{T} \left(\hat{x}_{0|t}^i - \hat{x}_{0|t}^{i,\text{NN}}\right) \mathbb{I}\left\{\left\|\hat{x}_{0|t}^i - \hat{x}_{0|t}^{i,\text{NN}}\right\| < SD\right\}, \qquad (2)$$

where $\mathbb{I}\{\cdot\}$ is the indicator function.

## 6  EXPERIMENTS

We now evaluate the effectiveness of our guidance approach in enhancing meaningful diversity. We focus on four diffusion-based restoration methods that attempt to draw samples from the posterior: RePaint (Lugmayr et al., 2022a), DDRM (Kawar et al., 2022), DDNM (Wang et al., 2022), and DPS (Chung et al., 2023). For each of them, we compare the restorations generated by the vanilla method to those obtained with our guidance, using the same noise samples for a fair comparison. We experiment with inpainting and super-resolution as our restoration tasks, using the same datasets, images, and masks (where applicable) as in Sec. 4.1 (for results on image colorization see App. I.1). In addition to the task of noisy $16\times$ super-resolution on CelebAMask-HQ, we add noisy $4\times$ super-resolution on PartImagenet, as well as noiseless super-resolution on both datasets. We also experimented with the methods of (Choi et al., 2021; Wei et al., 2022) for noiseless super-resolution, but found their consistency to be poor (LR-PSNR $< 45$dB) and thus discarded them (see App. C). In all our experiments, we use $N = 5$ representatives to compose the set $\mathcal{X}$. We conduct quantitative comparisons, as well as user studies. Qualitative results are shown in Fig. 7 and in App. I.

**Human perception tests.** As in Sec. 4.1, we conducted a *paired diversity test* to compare between the vanilla generation and the diversity-guided generation. However, in contrast to Sec. 4.1, in which the restorations in $\mathcal{X}$ were chosen among model outputs, here we intervene in the generation process. We therefore also examined whether this causes a decrease in image quality, by conducting a *paired image quality test* in which users were asked to choose which one of two images has a higher quality: a vanilla restoration or a guided one. The studies share the base configuration used in Sec. 4.1, using RePaint for inpainting and DDRM for super-resolution, both on images from the CelebAMask-HQ dataset (see App. E for additional details). As seen in Fig. 8, the guided restorations were chosen as more diverse significantly more often, while their quality was perceived as at least comparable.

Table 1: Quantitative results on CelebAMask-HQ in noisy (left) and noiseless (right) 16× super resolution. For each method we report results of both vanilla sampling and sampling with guidance.

| Model | $\sigma = 0.05$ | | | $\sigma = 0$ | | |
|---|---|---|---|---|---|---|
| | LPIPS Div. (↑) | NIQE (↓) | LR-PSNR | LPIPS Div. (↑) | NIQE (↓) | LR-PSNR (↑) |
| DDRM | 0.19 | 8.47 | 31.59 | 0.18 | 8.30 | 54.69 |
| + Guidance | **0.25** | 7.85 | 31.00 | **0.24** | 7.54 | 53.82 |
| DDNM | N/A | N/A | N/A | 0.18 | 7.40 | 81.24 |
| + Guidance | N/A | N/A | N/A | **0.26** | 6.92 | 75.04 |
| DPS | 0.29 | 5.72 | 30.00 | 0.25 | 5.41 | 52.05 |
| + Guidance | **0.34** | 5.16 | 28.98 | **0.28** | 5.05 | 53.45 |

Table 2: Quantitative results on PartImageNet in noisy (left) and noiseless (right) 4× super resolution. For each method we report results of both vanilla sampling and sampling with guidance.

| Model | $\sigma = 0.05$ | | | $\sigma = 0$ | | |
|---|---|---|---|---|---|---|
| | LPIPS Div. (↑) | NIQE (↓) | LR-PSNR | LPIPS Div. (↑) | NIQE (↓) | LR-PSNR (↑) |
| DDRM | 0.15 | 10.16 | 32.86 | 0.09 | 8.93 | 55.36 |
| + Guidance | **0.18** | 9.06 | 32.80 | **0.12** | 8.18 | 55.12 |
| DDNM | N/A | N/A | N/A | 0.10 | 9.63 | 71.40 |
| + Guidance | N/A | N/A | N/A | **0.16** | 8.52 | 69.54 |
| DPS | 0.31 | 7.27 | 28.25 | 0.33 | 15.27 | 46.92 |
| + Guidance | **0.33** | 7.62 | 28.06 | **0.40** | 20.48 | 47.56 |

**Quantitative analysis.** Tables 1, 2 and 3 report quantitative comparisons between vanilla and guided restoration. As common in diverse restoration works (Alkobi et al., 2023; Saharia et al., 2022; Zhao et al., 2021), we use the average LPIPS distance (Zhang et al., 2018) between all pairs within the set as a measure for semantic diversity.

We further report the NIQE image quality score (Mittal et al., 2012), and the LR-PSNR metric which quantifies consistency with the low-resolution input image in the case of super-resolution. We use N/A to denote configurations that are missing in the source codes of the base methods. For comparison, we also report the results of the GAN based inpainting method MAT (Li et al., 2022), which achieves lower diversity than all diffusion-based methods. As seen in all tables, our guidance method improves the LPIPS diversity while maintaining similar NIQE and LR-PSNR levels. The only exception is noiseless inpainting with DPS in Tab. 2, where the NIQE increases but is poor to begin with.

Table 3: Quantitative results on CelebAMask-HQ (left) and PartImageNet (right) in image inpainting.

| Model | CelebAMask-HQ | | PartImageNet | |
|---|---|---|---|---|
| | LPIPS Div. (↑) | NIQE (↓) | LPIPS Div. (↑) | NIQE (↓) |
| MAT | 0.03 | 4.69 | N/A | N/A |
| DDNM | 0.06 | 5.35 | 0.07 | 5.82 |
| + Guidance | **0.09** | 5.31 | **0.08** | 5.81 |
| RePaint | 0.08 | 5.07 | 0.09 | 5.41 |
| + Guidance | **0.09** | 5.05 | **0.10** | 5.34 |
| DPS | 0.07 | 4.97 | 0.10 | 5.35 |
| + Guidance | **0.09** | 4.91 | **0.11** | 5.37 |

# 7 CONCLUSION

We showed that posterior sampling, a strategy that has gained popularity in image restoration, is limited in its ability to summarize the range of semantically different solutions with a small number of samples. We thus proposed to break-away from posterior sampling and rather aim for composing small but meaningfully diverse sets of solutions. We started by a thorough exploration of what makes a set of reconstructions meaningfully diverse, and then harnessed the conclusions for developing diffusion-based restoration methods. We demonstrated quantitatively and via user studies that our methods outperform vanilla posterior sampling. Directions for future work are outlined in App. H.

## ETHICS STATEMENT

As the field of deep learning advances, image restoration models find increasing use in the everyday lives of many around the globe. The ill-posed nature of image restoration tasks, namely, the lack of a unique solution, contribute to uncertainty in the results of image restoration. This is especially crucial when the use is for scientific imaging, medical imaging, and other safety critical domains, where presenting restorations that are all drawn from the dominant modes may lead to misjudgements regarding the true, yet unknown, information in the original image. It is thus important to outline and visualize this uncertainty when proposing restoration methods, and to convey to the user the abundance of possible solutions. We therefor believe that the discussed concept of meaningfully diverse sampling could benefit the field of image restoration, commencing with the proposed approach.

## REPRODUCIBILITY STATEMENT

We refer to our code repository from our project's webpage at [https://noa-cohen.github.io/MeaningfulDiversityInIR/](https://noa-cohen.github.io/MeaningfulDiversityInIR/). The repository includes the required scripts for running all of the proposed baseline approaches, as well as code that includes guidance for all four image restoration methods compared in the paper.

## ACKNOWLEDGEMENTS

The research of TM was partially supported by the Israel Science Foundation (grant no. 2318/22), by the Ollendorff Miverva Center, ECE faculty, Technion, and by a gift from KLA. YB has received funding from the European Union's Horizon 2020 research and innovation programme under the Marie Skłodowska-Curie grant agreement no. 945422. The Miriam and Aaron Gutwirth Memorial Fellowship supported the research of HM.

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

# SUPPLEMENTARY MATERIAL

## A    DETAILS OF THE UNIFORMIZATION APPROACH

Let $f(x_i)$ denote the probability density function of the posterior[1] at point $x_i$, and assume for now that it is known and has a compact support. In the Uniformization method we assign to each member of $\mathcal{X}$ a probability mass that is inversely proportional to its density,

$$W(x_i) = \frac{\frac{1}{f(x_i)}}{\sum_{j=1}^{M} \frac{1}{f(x_j)}}. \tag{3}$$

We then populate $\mathcal{X}$ by sampling from $\tilde{\mathcal{X}}$ without repetition according to the probabilities $W(x_i)$.

In practice, the probability density $f(x)$ is not known. We therefore estimate it from the samples in $\tilde{\mathcal{X}}$ using the $k$-Nearest Neighbor (KNN) density estimator (Zhao & Lai, 2022),

$$\hat{f}(x) = \frac{k - 1}{M \cdot V(\mathcal{B}(\rho_k(x)))}. \tag{4}$$

Here, $\rho_k(x)$ is the distance between $x$ and its $k^{\text{th}}$ nearest neighbor in $\tilde{\mathcal{X}}$ and $V(\mathcal{B}(r))$ is the volume of a ball of radius $r$, which in $d$-dimensional Euclidean space is given by

$$V(\mathcal{B}(r)) = \frac{\pi^{d/2}}{\Gamma\left(\frac{d}{2} + 1\right)} r^d, \tag{5}$$

where $\Gamma$ is the gamma function. Using equation 5 in equation 4 and substituting the result into equation 3, we finally obtain the sampling probabilities for $x_i \in \tilde{\mathcal{X}}$:

$$W(x_i) = \frac{\rho_k(x_i)^d}{\sum_{j=1}^{M} \rho_k(x_j)^d}. \tag{6}$$

Note that in many cases, the support of $P_{X|Y}$ may be very large, or even unbounded. This implies that the larger our initial set $\tilde{\mathcal{X}}$ is, the larger the chances that it includes highly unlikely and peculiar solutions. Although these are in principle valid restorations, below some degree of likelihood, they are not representative of the *effective support* of the posterior. Hence, we omit the $\tau$ percent of the least probable restorations.

A more inherent limitation of the Uniformization method is that it may under-represent high-probability modes if their effective support is small. This can be seen in Fig. 4, where although Uniformization leads to a diverse set, this set does not contain a single representative from the dominant mode. Thus in this case, $95\%$ of the samples in $\tilde{\mathcal{X}}$ do not have a single representative in $\mathcal{X}$.

Note that estimating distributions in high dimensions is a fundamentally difficult task, which requires the number of samples $N$ to be exponential in the dimension $d$ to guarantee reasonable accuracy. This problem can be partially resolved by reducing the dimensionality of the features (*e.g.*, using PCA) prior to invoking the KNN estimator. We follow this practice in all our feature based image restoration experiments.

We used $\tau = 100\%$, $k = 10$ for the estimator in the toy example, and $k = 6$ otherwise.

---

[1]For notational convenience we omit the dependence on $y$.

# B   The Effect of discarding points from $\tilde{\mathcal{X}}$ in the baseline approaches

With all subsampling approaches, we exploit distances calculated over a semantic feature space as means for locating interesting representative images. Specifically, we consider an image that is far from all other images in our feature space as being semantically dissimilar to them, and the general logic we discussed thus far is that such an image constitutes an interesting restoration solution and thus a good candidate to serve as a representative sample. However, beyond some distance, dissimilarity to all other images may rather indicate that this restoration is unnatural and too improbable, and thus would better be disregarded.

Two of our analyzed subsampling approaches address this aspect using an additional hyperparameter. In FPS, we denote by $L$ the number of solutions that are randomly sampled from $\tilde{\mathcal{X}}$ before initiating the FPS algorithm. Due to the heavy tail nature of the posterior, setting a smaller value for $L$ increases the chances of discarding such unnatural restorations even before running FPS, which in turn results in a subset $\mathcal{X}$ leaning towards the more likely restorations. Similarly, in the Uniformization approach, we can use only a certain percent $\tau$ of the solutions. However, while the $L$ solutions that we keep in the FPS case are chosen randomly, here we take into account the estimated probability of each solution in order to intentionally use only the $\tau$ percent of the most probable restorations. Figures 9 and 10 illustrate the effects of those parameters in the toy Gaussian mixture problem discussed in the main text, while Figures 11-13 illustrate their effect in image restoration experiments.

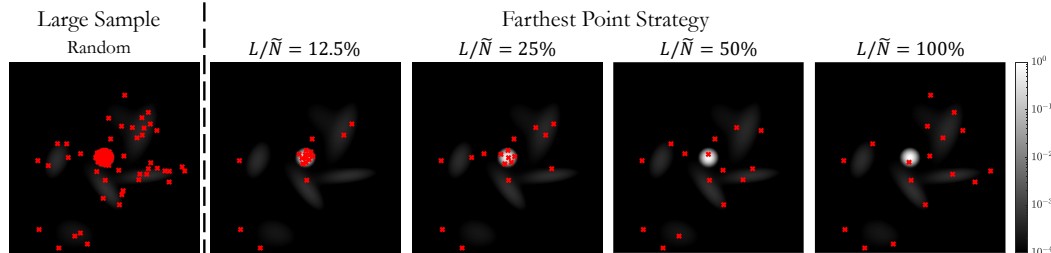

Figure 9: **Effect of $L$ on FPS sampling**. A toy example comparing the represented set sampled from an imbalanced mixture of 10 Gaussians (left), using a subset $\mathcal{X}$ of only $N = 20$ points, for different values of $L$. Note how the samples spread as $L$ approaches $\tilde{N}$.

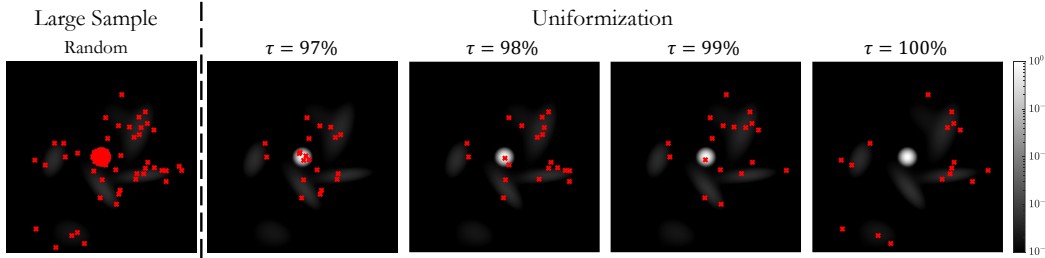

Figure 10: **Effect of $\tau$ on Uniformization sampling**. A toy example comparing the represented set sampled from an imbalanced mixture of 10 Gaussians (left), using a subset $\mathcal{X}$ of only $N = 20$ points, for different values of $\tau$. Note how the central Gaussian which contains 95% of the probability mass contains no samples for $\tau = 100\%$.

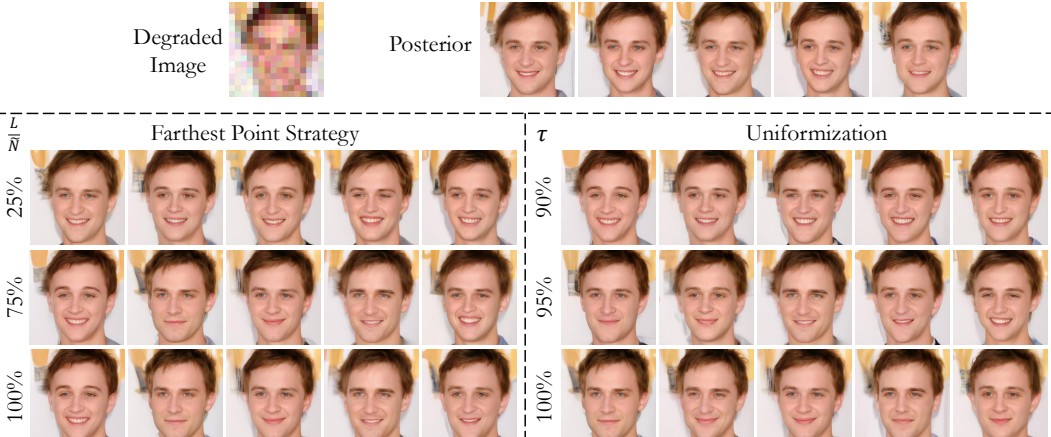

Figure 11: **Effect of discarding points before subsampling in super-resolution on CelebAMask-HQ.** Note the lack in representation of open-mouth smiles when applying Uniformization (right) with $\tau = 100\%$, despite the fact that smiles dominate the approximated posterior distribution. This aligns with the behaviour of the toy example.

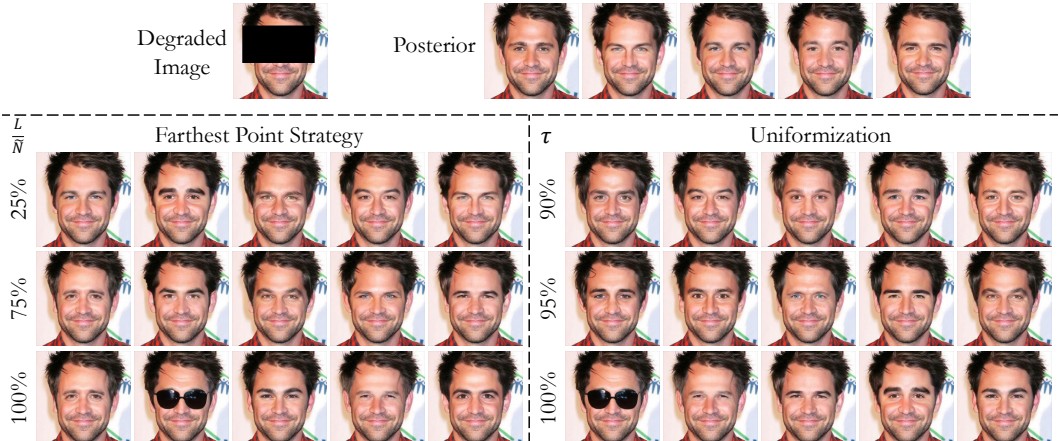

Figure 12: **Effect of discarding points before subsampling in image inpainting on CelebAMask-HQ.** Note how the sunglasses inpainting option is among the first to be omitted in both subsampling methods. This demonstrates the effect of hyper-parameters $L$ and $\tau$ on the maximal degree of presented peculiarity in the FPS and Uniformization approaches, respectively.

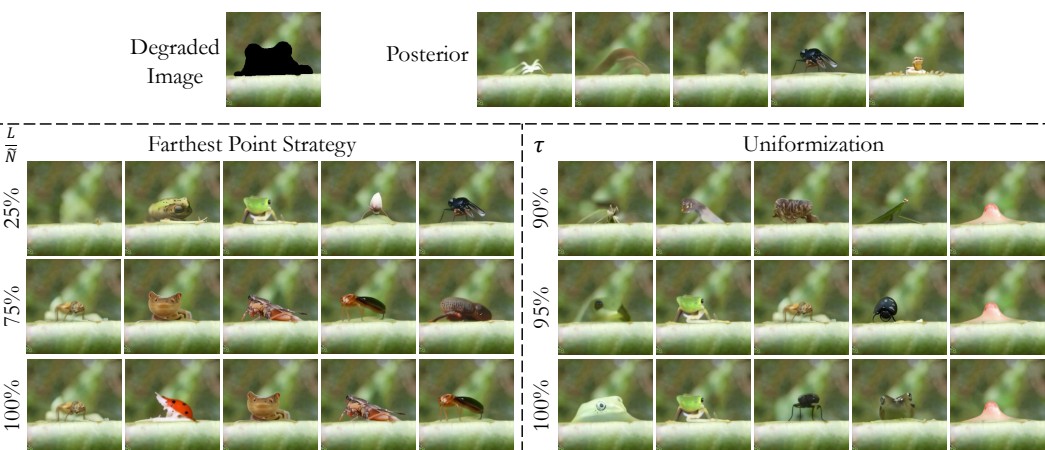

Figure 13: **Effect of discarding points before subsampling in image inpainting on PartImagenet.**

## C  EXPERIMENTAL DETAILS

**Pre-processing.**  In all experiments, we crop the images into a square and resize them to $256 \times 256$, to satisfy the input dimensions expected by all models. For all super-resolution experiments, bicubic downsampling was applied to the original images to create their degraded version, and random noise was added for noisy super-resolution (according to the denoted noise level).

**Masks.**  For face inpainting, we use the landmark-aligned face images in the CelebAMask-HQ dataset and define four masks: Large and small masks covering roughly the area of the eyes, and large and small masks covering the mouth and chin. For each face we sample one of the four possible masks. For inpainting of PartImagenet images, we combine the masks of all parts of the object and use the minimal bounding box that contains them all.

**Pretrained models.**  For PartImageNet we use the same checkpoint of Dhariwal & Nichol (2021) across all models. For CelebAMask-HQ, we use the checkpoint of Meng et al. (2022) in DDRM, DDNM and DPS, and in RePaint we use the checkpoint used in their source code.

**Guidance parameters.**  The varied diffusion methods used in all guidance experiments (Chung et al., 2023; Kawar et al., 2022; Lugmayr et al., 2022a; Wang et al., 2022) display noise spaces with different statistics during their sampling process. This raises the need for differently tuned guidance hyper-parameters for each method, and sometimes for different domains. Tabs. 4 and 5 lists the guidance parameters used in all figures and tables presented in the paper.

Table 4: Values of the guidance step-size hyper-parameter $\eta$ used in our experiments.

| Model | CelebAMask-HQ | | | PartImageNet | | |
|---|---|---|---|---|---|---|
| | Super Resolution | | Image Inpainting | Super Resolution | | Image Inpainting |
| | $\sigma = 0.05$ | $\sigma = 0$ | | $\sigma = 0.05$ | $\sigma = 0$ | |
| DDRM | 0.8 | 0.8 | N/A | 0.8 | 0.8 | N/A |
| DDNM | N/A | 0.8 | 0.9 | N/A | 0.8 | 0.9 |
| DPS | 0.5 | 0.3 | 0.5 | 0.5 | 0.5 | 0.5 |
| RePaint | N/A | N/A | 0.3 | N/A | N/A | 0.3 |

Table 5: Values of $D$, *i.e.*, the upper distance bound for applying guidance, used in our experiments.

| Model | CelebAMask-HQ | | | PartImageNet | | |
|---|---|---|---|---|---|---|
| | Super Resolution | | Image Inpainting | Super Resolution | | Image Inpainting |
| | $\sigma = 0.05$ | $\sigma = 0$ | | $\sigma = 0.05$ | $\sigma = 0$ | |
| DDRM | 0.0004 | 0.0004 | N/A | 0.0004 | 0.0004 | N/A |
| DDNM | N/A | 0.0005 | 0.0015 | N/A | 0.0003 | 0.0008 |
| DPS | 0.06 | 0.001 | 0.009 | 0.0005 | 0.0005 | 0.009 |
| RePaint | N/A | N/A | 0.00028 | N/A | N/A | 0.00028 |

**Consistency constraints.**  As explained in Sec. 6, in our experiments we consider only consistent models. We regard a model as consistent based on the mean PSNR of its reconstructions computed between the degraded input image $y$ and a degraded version of the reconstruction, *e.g.*, LR-PSNR in the task of super-resolution. For noiseless inverse problems, such as noiseless super-resolution or inpainting, we follow Lugmayr et al. (2022b) and use 45dB as the minimal required value to be considered consistent. This allows for a deviation of a bit more than one gray-scale level. We additionally experimented with ILVR (Choi et al., 2021) and E2Style (Wei et al., 2022), which were both found to be inconsistent (even when trying to tune ILVR's range hyper-parameter; E2Style does not have a parameter to tune).

## DPS Vanilla Generations

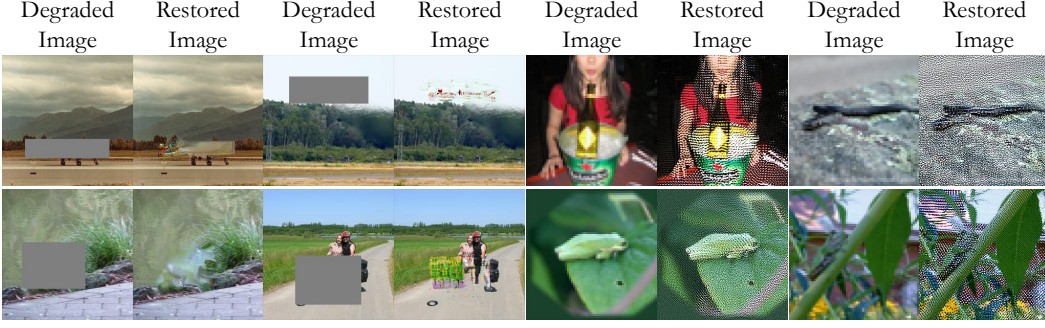

Figure 14: **Examples of artifacts in the generations of DPS.** We show here results generated by DPS with $\zeta_i = 2$ for inpainting, and $\zeta_i = 10$ for noiseless super-resolution.

**Tuning the hyper-parameter $\zeta_i$ of DPS.** While RePaint (Lugmayr et al., 2022a), MAT (Li et al., 2022), DDRM (Kawar et al., 2022) and DDNM (Wang et al., 2022) are inherently consistent, DPS (Chung et al., 2023) is not, and its consistency is controlled by a $\zeta_i$ hyper-parameter. To allow fair comparison, we searched for the minimal $\zeta_i$ value per experiment, that would yield consistent results without introducing saturation artifacts (which we found to increase with $\zeta_i$). In particular, for inpainting on CelebAMask-HQ we use $\zeta_i = 2$. Since we were unable to find $\zeta_i$ values yielding consistent and plausible (artifact-free) results for inpainting on PartImageNet, we resorted to using the setting of $\zeta_i = 2$ adopted from the the CelebAMask-HQ configuration, when reporting the results in the bottom right cell of Tab. 3. However, note that the restorations there are inconsistent with their corresponding inputs. For noiseless super-resolution on CelebAMask-HQ we use $\zeta_i = 10$. For noiseless super-resolution on PartImageNet we use $\zeta_i = 3$. While this is the minimal value that we found to yield consistent results, these contained saturation artifacts, as evident by their corresponding high NIQE value in Tab. 2. We nevertheless provide the results for completeness. For noisy super-resolution, we follow a rule of thumb of having the samples' LR-PSNR around 26dB, which aligns with the expectation that the low-resolution restoration should deviate from the low-resolution noisy input $y$ by approximately the noise level $\sigma_y = 0.05$. Following this rule of thumb we use $\zeta_i = 1$ for noisy super-resolution on both domains. Examples of DPS saturation artifacts (resulting from high $\zeta_i$ values) can be seen in Fig. 14.

**Measured diversity and image quality.** In Tabs. 1, 2 and 3 we report LPIPS diversity and NIQE[2]. In all experiments, the LPIPS diversity was computed by measuring the average LPIPS distance over all possible pairs in $\mathcal{X}$, with VGG-16 (Simonyan & Zisserman, 2015) as the neural features architecture.

---

[2]We use PyTorch Toolbox for Image Quality Assessment available at https://github.com/chaofengc/IQA-PyTorch for computing NIQE and the LPIPS distance.

# D   THE EFFECTS OF THE GUIDANCE HYPERPARAMETERS

Here we discuss the effects of the guidance hyperparameters $\eta$, which is the step size that controls the guidance strength, and $D$, which controls the minimal distance from which we do not apply a guidance step.

We provide qualitative and quantitative results on CelebAMask-HQ image inpainting in Figs. 15 and 16 and Tabs. 6 and 7, respectively. As can be seen, increasing $\eta$ yields higher diversity in the sampled set. Too large $\eta$ values can cause saturation effects, but those can be at least partially mitigated by adjusting $D$ accordingly. Intuitively speaking, increasing $D$ allows for larger changes to take effect via guidance. This means that the diversity increases as $D$ increases. The effect of using a minimal distance $D$ at all is very noticeable in the last row of Fig. 16. In this row, no minimal distance was set, and therefore the full guidance strength of $\eta$ is visible. Setting $D$ allows truncating the effect for some samples, while still using a large $\eta$ that will help push similar samples away from one another. Both hyperparameters work in conjunction, as setting one parameter too-small will yield to lower diversity.

Table 6: Effect of $\eta$ on results for CelebAMash-HQ in image inpainting. Here $D$ is fixed at 0.0003.

| $\eta$ | LPIPS Div. (↑) | NIQE (↓) |
|---|---|---|
| 0 (Posterior) | 0.090 | 5.637 |
| 0.07 | 0.105 | 5.495 |
| 0.1 | 0.109 | 5.506 |
| 0.3 | 0.113 | 5.519 |

Table 7: Effect of $D$ on results for CelebAMash-HQ in image inpainting. Here $\eta$ is fixed at 0.09.

| $D$ | LPIPS Div. (↑) | NIQE (↓) |
|---|---|---|
| 0 | 0.090 | 5.637 |
| 0.0002 | 0.094 | 5.552 |
| 0.0003 | 0.108 | 5.472 |
| 0.0004 | 0.126 | 5.554 |
| $\infty$ | 0.141 | 5.450 |

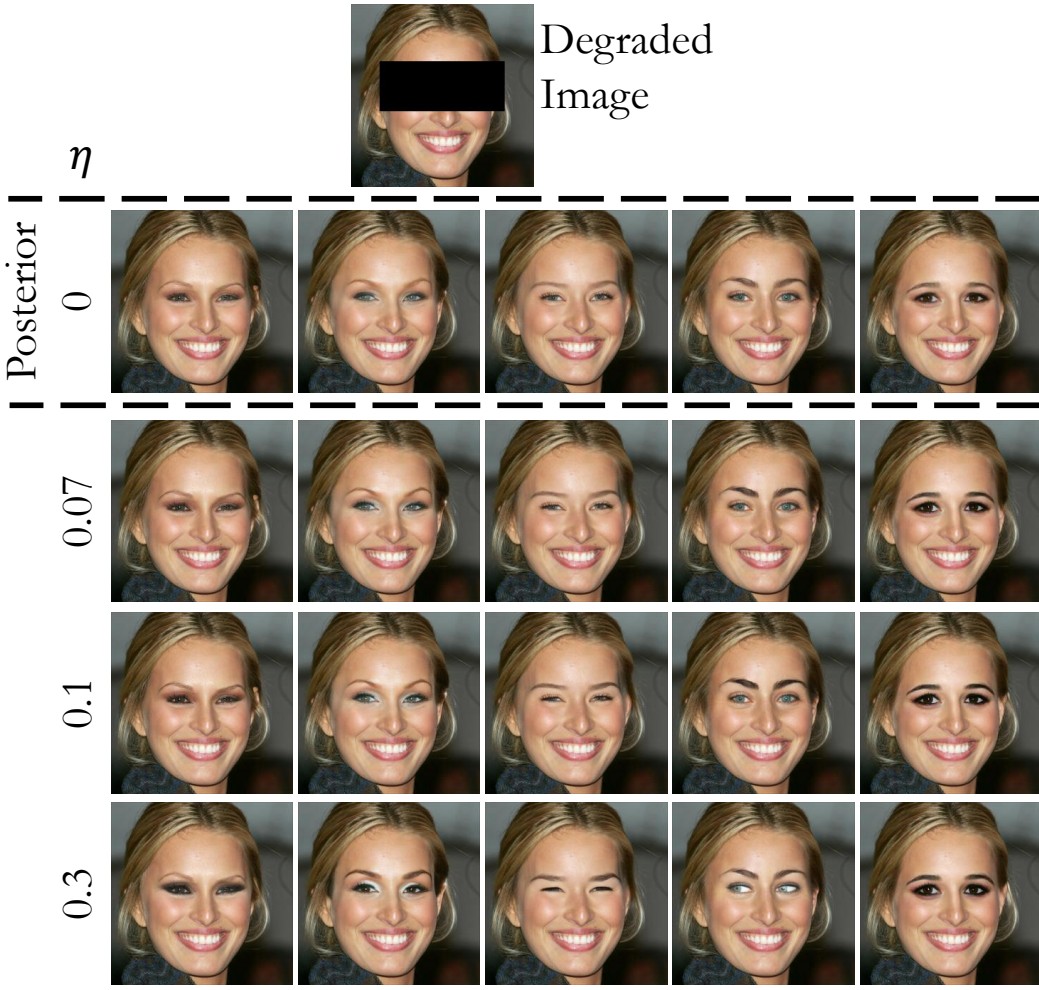

Figure 15: **The effect of using different step sizes $\eta$ on the diversity of the results.** Here, we fix $D$ to 0.003.

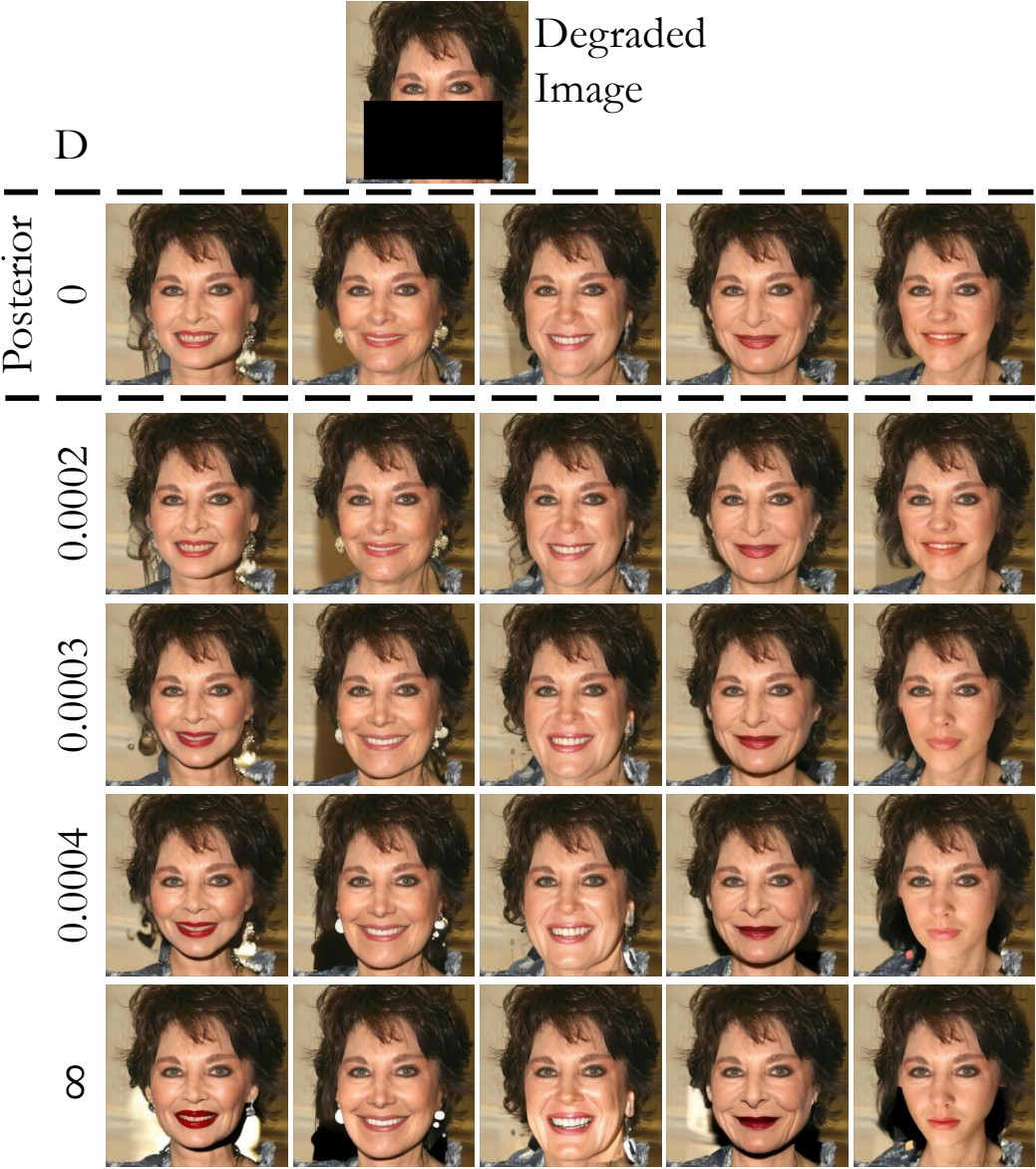

Figure 16: **The effect of using different upper bound distances $D$ on the diversity of the results.** In this example, not setting an upper bound (last row) results in some mild artifacts, e.g. overly bright regions as well as less-realistic earring appearances. Here, we fix $\eta$ to 0.09.

# E    USER STUDIES

Beyond reporting the results in Fig. 6 in the main text, we further visualize the data collected in the user studies discussed in 4.1 on a $2D$ plane depicting the trade off between the two characteristics of each sampling approach: (i) the diversity perceived by users compared with the diversity of random samples from the approximated posterior, and (ii) the coverage of more likely solutions by the sub-sampled set $\mathcal{X}$. All sub-sampling approaches achieve greater diversity compared to random samples from the approximate posterior in both super-resolution and inpainting tasks, performed on images from the CelebAMask-HQ dataset Lee et al. (2020). However, the visualization in Fig. 17 indicates their corresponding different positions on the diversity-coverage plane. In both tasks, sampling according to $K$-means achieves the highest coverage of likely solutions, at the expense of relatively low diversity values. Sub-sampling using FPS achieves the highest diversity.

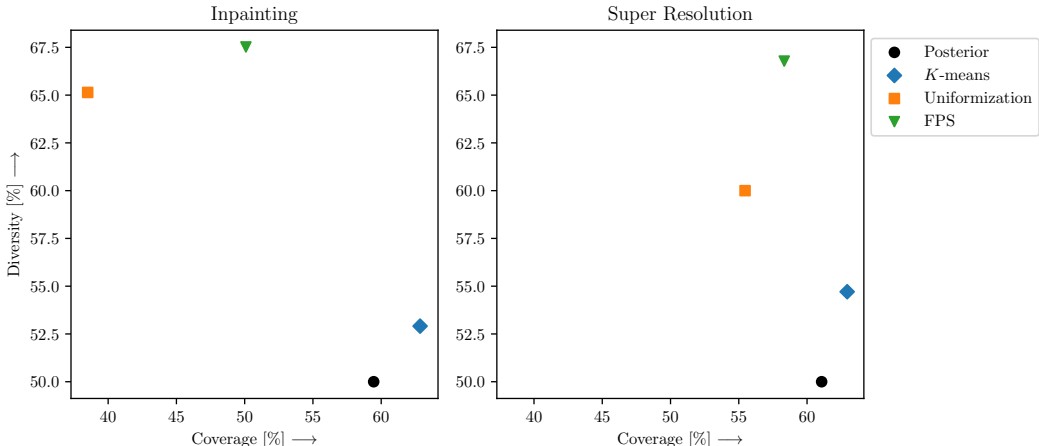

Figure 17: **Diversity-Coverage plane**. A representative set needs to trade-off covering the possible solution set, and seeking diversity in the subset of images presented. Diversity of the three explored approaches was measured relative to approximated posterior samples, hence the value determined for the posterior sampling is in theory $50\%$.

In our exploration of what mathematically characterizes a meaningfully diverse set of solutions in Sec. 4 we build upon semantic deep features. The choice of which semantic deep features to use in the sub-sampling procedure impacts the diversity as perceived by users, and should therefore be tuned according to the type of diversity aimed for. In the user studies of these baseline approaches, discussed in 4.1, we did not guide the users what type of diversity to seek (*e.g.*, diverse facial expressions vs. diverse identities). However, for all our sub-sampling approaches, we used deep features from the AnyCost attribute predictor Lin et al. (2021). We now validate our choice to use $L^2$ distance over such features as a proxy for human perceptual dissimilarity by comparing it with other feature domains and metrics in Fig. 18. Each plot depicts the distances from the target images presented in each question to their nearest neighbor amongst the set of images $\mathcal{X}$ presented to the user, against the percentage of users who perceived at least one of the presented images as similar to the target. We utilize a different feature domain to calculate the distances in each sub-plot, and report the corresponding Pearson and Spearman correlations in the sub-titles (lower is better, as we compare distance against similarity). All plots correspond to the image-inpainting task. Note that the best correlation is measured for the case of using the deep features of the attribute predictor, compared to using cosine distance between deep features of ArcFace Deng et al. (2019), using the pixels of the inpainted patches, or using the logits of the attribute predictor. The significant dimension reduction (*e.g.*, from 32768 to 25 dimensions in the case of the AnyCost attribute predictor) only slightly degrades correlation values when using distance over deep features. Finally, in Figs. 19-21 we include screenshots of the instructions and random questions presented to the users in all three types of user studies conducted in this work.

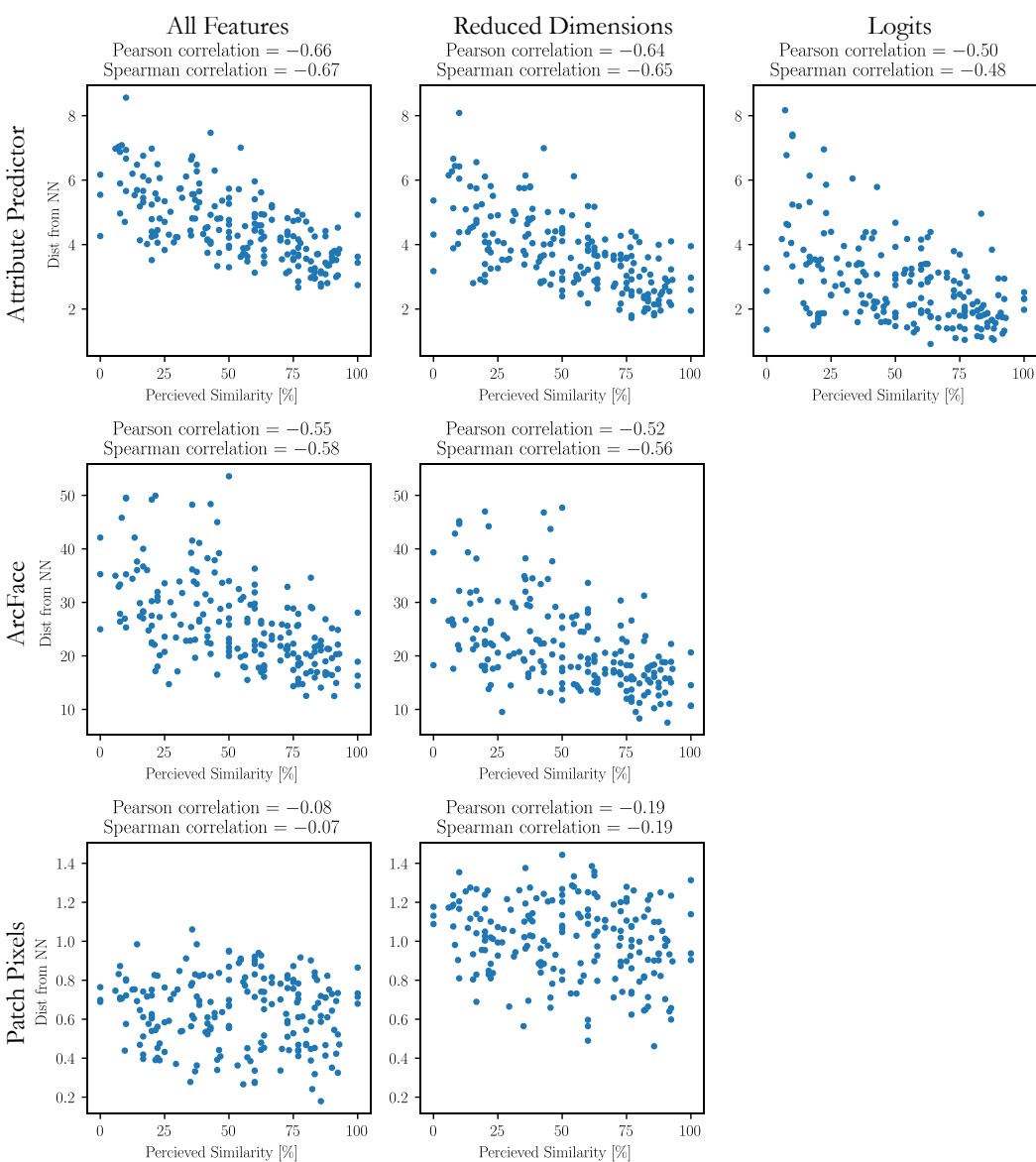

Figure 18: **Correlation between semantic distances and similarity as perceived by users**. Post processing of the data collected in the user study, evaluating the semantic distance used (top left).

## About this HIT:

- It should take about 6 minutes.

- This user study quantifies the effectiveness of new image restoration software.
- In each query, you'll see some image with a missing (black) region appearing at the top of the screen.
- Below the image, you will see 2 rows of 5 images. Select the row in which the images show a greater variety.

- You will complete a short practice (less than 1 minute) before starting.

Start the practice!

(a) Instructions presented to the user.

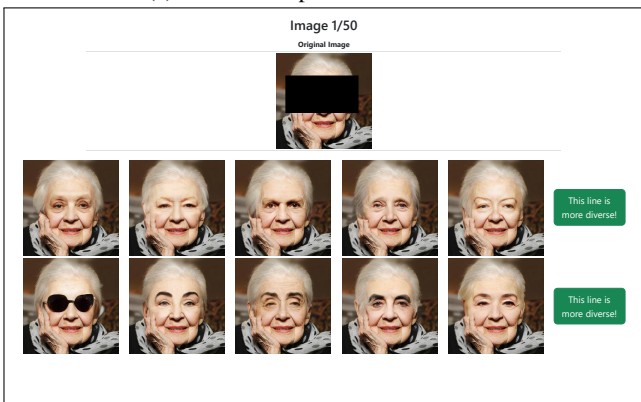

(b) Example for a question on a set of inpainting restorations.

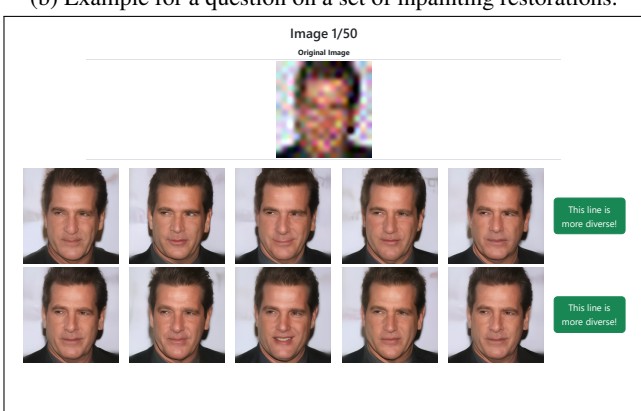

(c) Example for a question on a set of super resolution restorations.

Figure 19: **Paired diversity test**. After reading instructions (upper) participant had to answer which of the lines shows images with a greater variety.

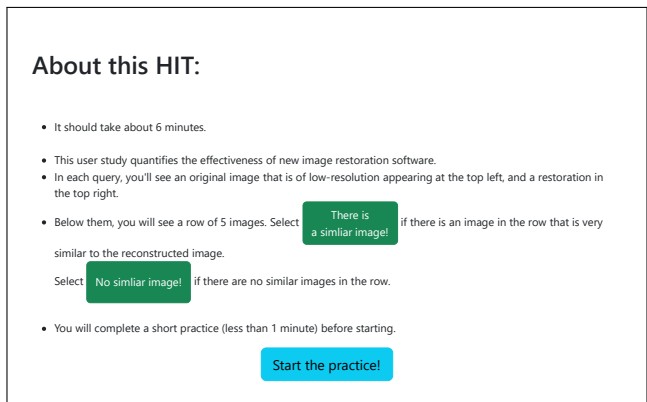

(a) Instructions presented to the user.

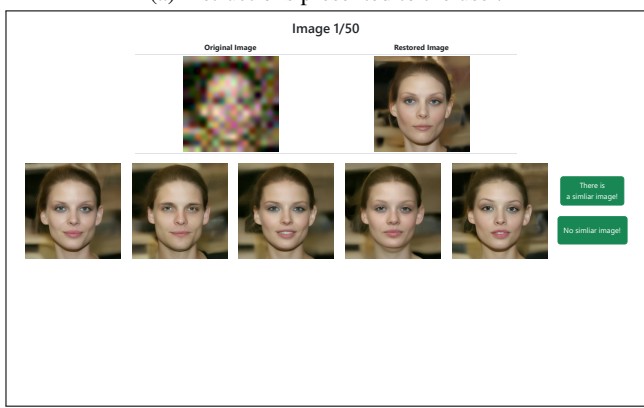

(b) Example for a question on a set of super resolution restorations.

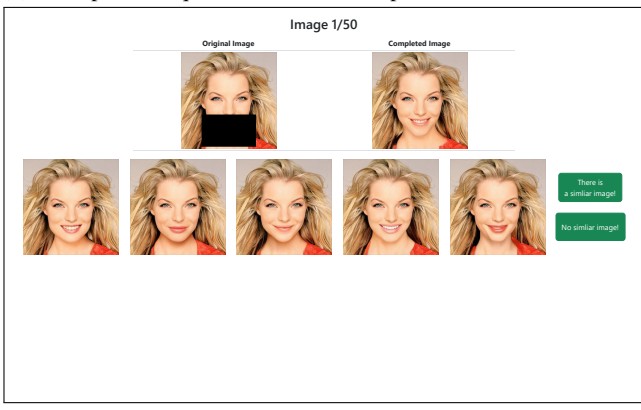

(c) Example for a question on a set of inpainting restorations.

Figure 20: **Unpaired coverage test**. After reading instructions (upper) participant had to answer whether any of the shown images is very similar to the target image.

## About this HIT:

- It should take about 2 minutes.

- This user study quantifies the effectiveness of new image restoration software.
- In each query, you'll see some low-resolution image appearing at the top of the screen.
- Below the image, you will see two restored high resolution versions of that image. Select the image which has a higher quality.

Start the experiment!

(a) Instructions presented to the user.

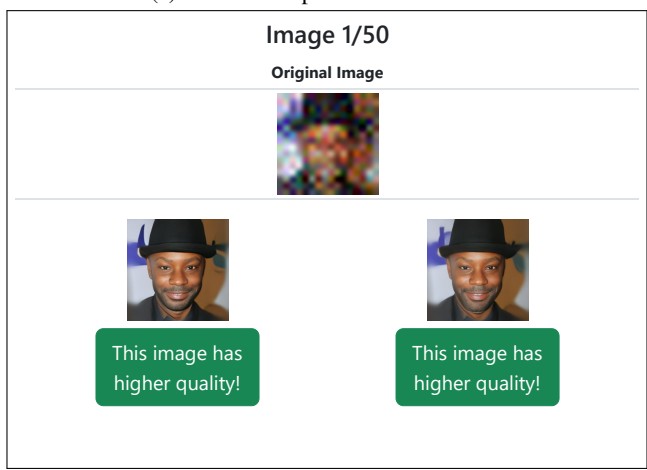

(b) Example for a question on a set of super resolution restorations.

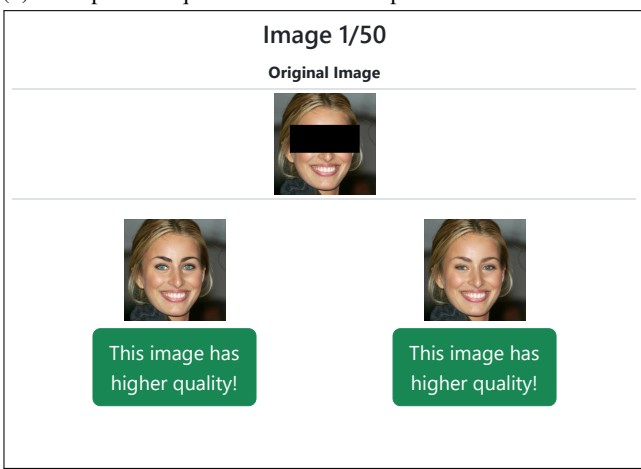

(c) Example for a question on a set of inpainting restorations.

Figure 21: **Paired image quality test**. After reading instructions (upper) participant had to answer which image is perceived with higher quality.

## F  AN ALTERNATIVE GUIDANCE STRATEGY

In Sec. 5 we proposed to increase the set's diversity by adding to each clean image prediction $\hat{x}_{0|t}^i \in \mathcal{X}_t$ the gradient of the dissimilarity between $\hat{x}_{0|t}^i$ and its nearest neighbor within the set,

$$\hat{x}_{0|t}^{i,\text{NN}} = \arg \min_{x \in \mathcal{X}_t \setminus \{\hat{x}_{0|t}^i\}} d(\hat{x}_{0|t}^i, x). \tag{7}$$

An alternative could be to use the dissimilarity between each image $\hat{x}_{0|t}^i$ and the average of all $N$ images in the set,

$$\hat{x}_{0|t}^{\text{AVG}} = \frac{1}{N} \sum_{j \in \{1,\dots,N\}} \hat{x}_{0|t}^j.$$

We opted for the simpler alternative that utilizes the nearest neighbor $\hat{x}_{0|t}^{i,\text{NN}}$, since we found the two alternatives to yield visually similar results. We illustrate the differences between the approaches in Figs. 22, 23, 24 and 25, where the values for $\eta$ in the inpainting task are 0.24 and 0.3 for the average and nearest neighbor cases respectively, and in the super resolution task are 0.64 and 0.8 for the average and nearest neighbor cases respectively.

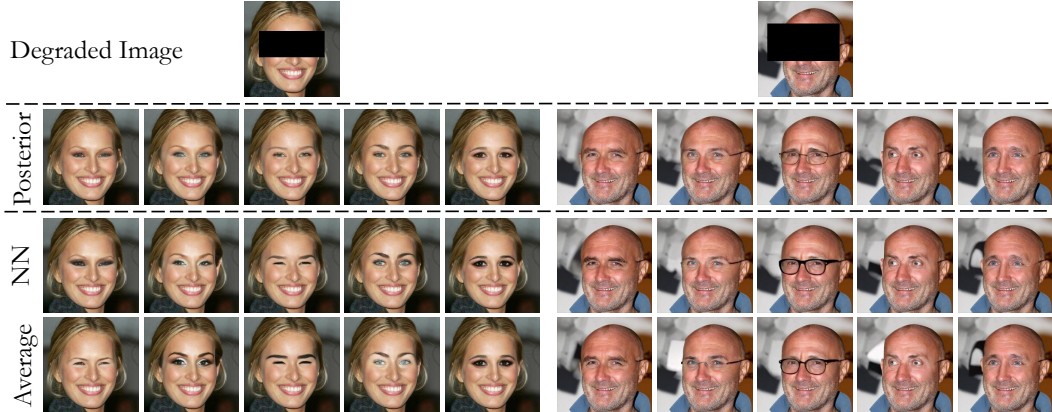

Figure 22: **Comparing the alternative guidance strategies for inpainting on CelebAMask-HQ.** We compare posterior sampling against both guidance using dissimilarity calculated relative to the nearest neighbor (NN) and against guidance using dissimilarity calculated relative to the average image of the set (Average). Restorations generated by RePaint (Lugmayr et al., 2022a).

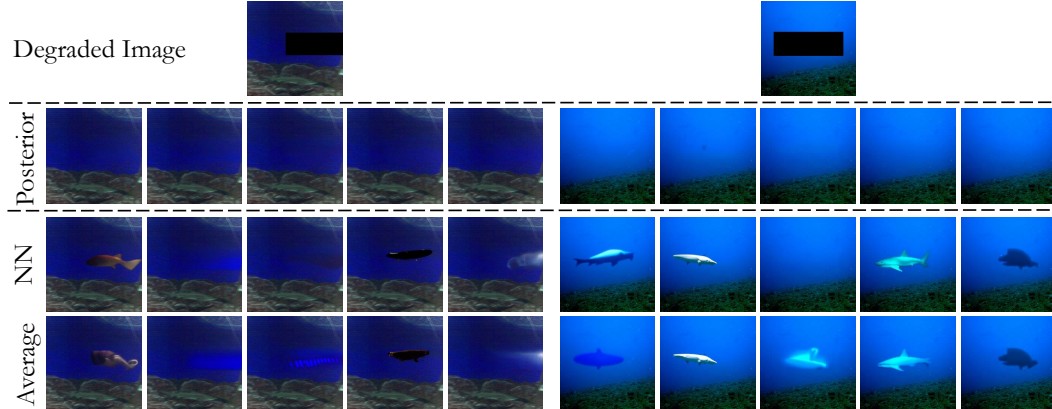

Figure 23: **Comparing the alternative guidance strategies for inpainting on PartImageNet.** We compare posterior sampling against both guidance using dissimilarity calculated relative to the nearest neighbor (NN) and against guidance using dissimilarity calculated relative to the average image of the set (Average). Restorations generated by RePaint (Lugmayr et al., 2022a).

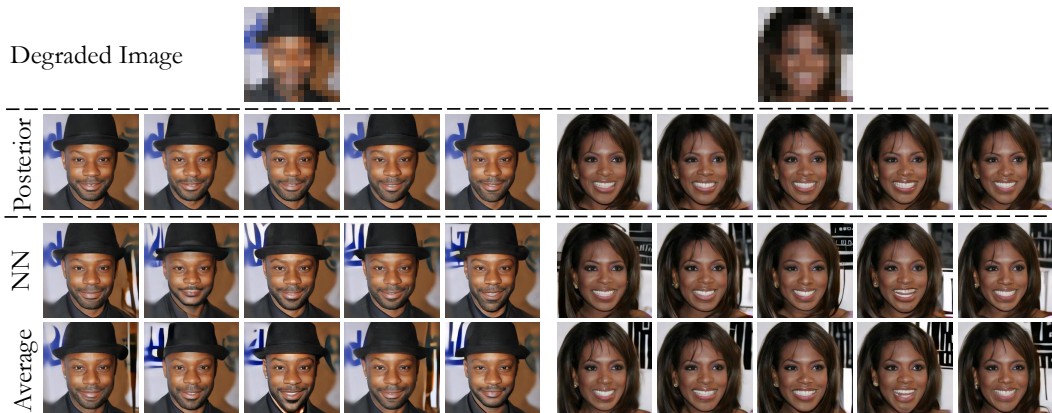

Figure 24: **Comparing the alternative guidance strategies for noiseless super-resolution on CelebAMask-HQ.** We compare posterior sampling against both guidance using dissimilarity calculated relative to the nearest neighbor (NN) and against guidance using dissimilarity calculated relative to the average image of the set (Average). Restorations generated by DDNM (Wang et al., 2022).

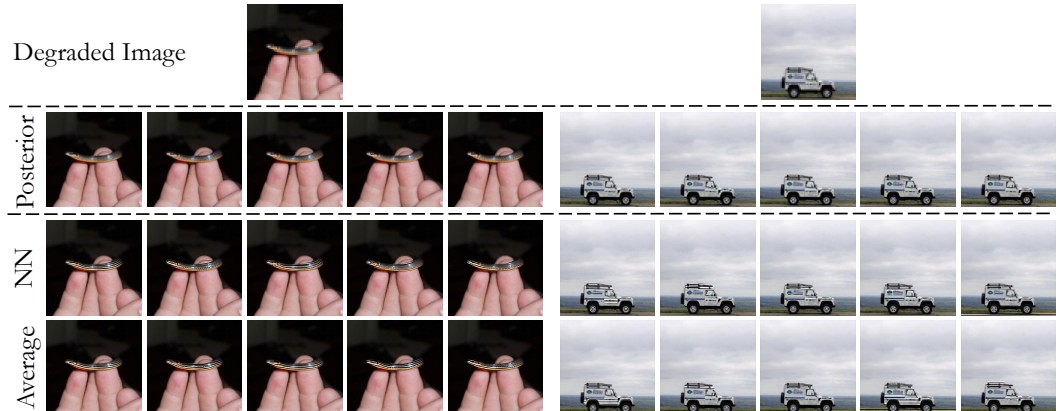

Figure 25: **Comparing the alternative guidance strategies for noiseless super-resolution on PartImageNet.** We compare posterior sampling against both guidance using dissimilarity calculated relative to the nearest neighbor (NN) and against guidance using dissimilarity calculated relative to the average image of the set (Average). Restorations generated by DDNM (Wang et al., 2022).

# G    HIERARCHICAL EXPLORATION

In some cases, a subset of $N$ restorations from $\tilde{\mathcal{X}}$ may not suffice for outlining the complete range of possibilities. A naive solution in such cases is to simply increase $N$. However, as $N$ grows, presenting a user with all images at once becomes ineffective and even impractical. We propose an alternative scheme to facilitate user exploration by introducing a hierarchical structure that allows users to explore the realm of possibilities in $\tilde{\mathcal{X}}$ in an intuitive manner, by viewing only up to $N$ images at time. This is achieved by organizing the restorations in $\tilde{\mathcal{X}}$ in a tree-like structure with progressively fine-grained distinctions. To this end, we exploit the semantic significance of the distances in feature space twice; once for sampling the representative set at each hierarchy level (*e.g.*, using FPS), and a second time for determining the descendants for each shown sample, by using its 'perceptual' nearest neighbor. This tree structure allows convenient interactive exploration of the set $\tilde{\mathcal{X}}$, where, at each stage of the exploration all images of the current hierarchy are presented to the user. Then, to further explore possibilities that are semantically similar to one of the shown images, the user can choose that image and move on to examine its children.

Constructing the tree consists of two stages that are repeated in a recursive manner: choosing representative samples from within a set of possible solutions (which is initialized as the whole set $\tilde{\mathcal{X}}$), and associating with each representative image a subset of the set of possible solutions which will form its own set of possible solutions down the recursion. Specifically, for each set of images not yet explored (initially all image restorations in $\tilde{\mathcal{X}}$), a sampling method is invoked to sample up to $N$ images and present them to the user. Any sampling method can be used (*e.g.*, FPS, Uniformization, etc.), which aims for a meaningful representation. All remaining images are associated with one or more of the sampled images based on similarity. This forms (up to) $N$ sets, each associated with one representative image. Now, for each of these sets, the process is repeated recursively until the associated set is smaller than $N$. This induces a tree structure on $\tilde{\mathcal{X}}$, demonstrated in Fig. 26, where the association was done by partitioning according to nearest neighbors by the similarity distance.

---

**Algorithm 1** Hierarchical exploration

---

$\tilde{\mathcal{X}}$: set of restored images
$N$: number of images to display at each time-step
SM: sampling method
PM: partition method

1: **function** MAIN
2:     MainRoot ← empty node
3:     EXPLOREIMAGES($\mathcal{X}$, MainRoot)

4: **function** EXPLOREIMAGES(images, root)
5:     **if** number of images $\leq N$ **then**
6:         root.children = images
7:     **else**
8:         children ← SM(images)
9:         **for** i $\in \{1, \cdots, N\}$ **do**
10:             Descendants ← PM(children, i, images)
11:             sub_tree ← EXPLOREIMAGES(Descendants, children[i])
12:             root.children.append(sub_tree)
13:     **return** root
**Ensure:** Tree data structure under MainRoot with all images as its vertices.

---

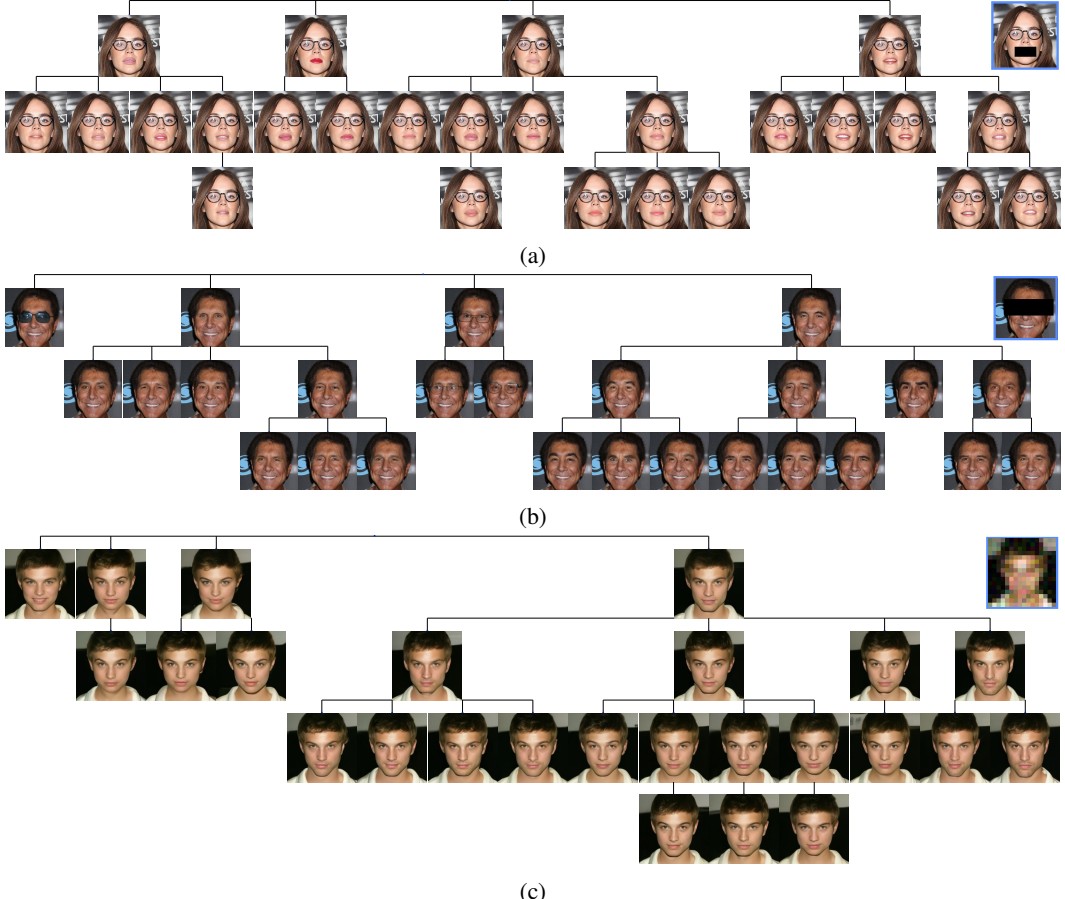

Figure 26: **Visualization of the hierarchical exploration**. The implied trees when setting $N = 4$, using the FPS sampling method, from a total of $\tilde{N} = 25$ restorations. The degraded image is marked in blue at the top right. Different attributes from the attribute predictor are expressed in each example, depending on the variety in the restorations. Note the variations in makeup and smile on the top pane, eye-wear and eyebrow expression on the middle pane and general appearance in the bottom pane.

## H    DIRECTIONS FOR FUTURE WORK

We investigated general meaningful diversity, which focuses on exploring different kinds of diversity at once. For example, in the context of restoration of face images, we aimed for our representative set to cover diverse face structures, glasses, makeup, etc. However, for certain applications it can be desirable to reflect the diversity for a specific property, *e.g.*, covering multiple types of facial hair and accessories while keeping the identity fixed, or covering multiple identities while keeping the facial expression fixed. The ability to achieve diversity in only specific attributes can potentially be important in *e.g.*, the medical domain, for example to allow a radiologist to view a range of plausible pathological interpretations for a specific tumor in a CT scan, or to present a forensic investigator with a representative subset of headwear that are consistent with a low quality surveillance camera footage. Additionally, We believe future work may focus on developing similar approaches for enhancing meaningful diversity in other restoration tasks.

## I    ADDITIONAL RESULTS

Throughout our experiments in the main paper we explored meaningful diversity in the context of inpainting and image super-resolution (with and without additive noise). To demonstrate the wide applicability of our method, we now present results on the task of image colorization, as well as additional comparisons on image inpainting and super-resolution.

### I.1    IMAGE COLORIZATION

Figures 27 and 28 present comparisons between our diversity guided generation and vanilla generation for the task of colorization. These results were generated using DDNM Wang et al. (2022), where the guidance parameters used for colorization on CelebAMask-HQ are $\eta = 0.08, D = 0.0005$, and the guidance parameters for colorization on PartImageNet are $\eta = 0.08, D = 0.0003$. As can be seen, our method significantly increases the diversity of the restorations, revealing variations in background and hair color, as well as in face skin tones. This is achieved while remaining consistent with the grayscale input images. Indeed, the average PSNR between the grayscale input image and the grayscale version of our reconstructions is 56.3dB for the face colorizations and 58.0dB for the colorizations of the PartImageNet images.

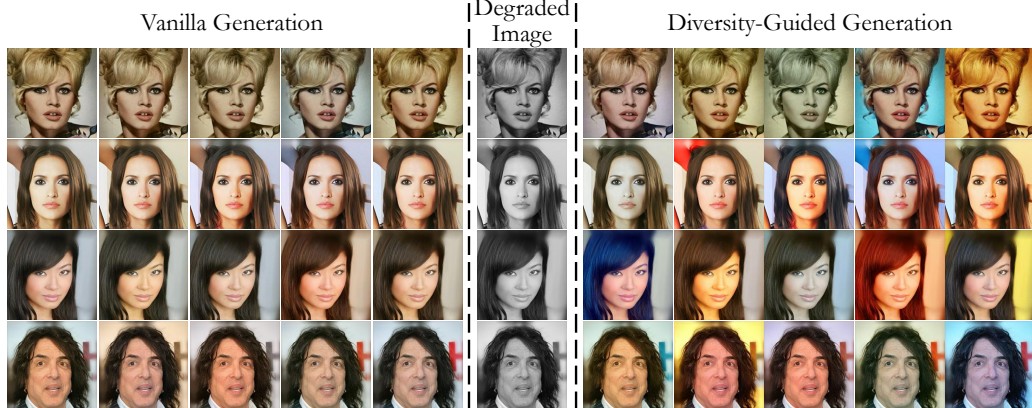

Figure 27: **Comparisons of colorization on CelebAMask-HQ, with and without diversity-guidance**. Restorations generated by DDNM (Wang et al., 2022).

Vanilla Generation      Degraded Image      Diversity-Guided Generation

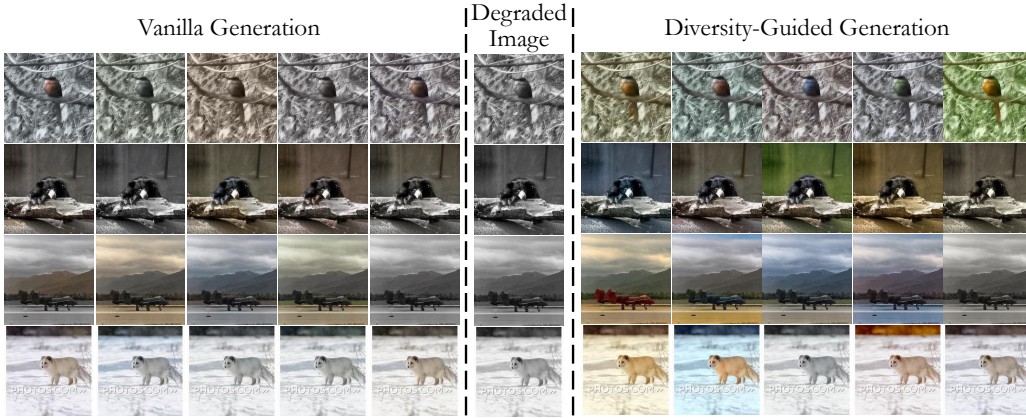

Figure 28: **Comparisons of colorization on PartImageNet, with and without diversity-guidance**. Restorations generated by DDNM (Wang et al., 2022).

## I.2 ADDITIONAL COMPARISONS ON IMAGE INPAINTING AND SUPER-RESOLUTION

Figure 29: **Additional comparisons of noisy** $16\times$ **super resolution with** $\sigma = 0.05$ **on CelebAMask-HQ with sub-sampling approaches vs. using the approximate posterior**. Restorations created using DDRM Kawar et al. (2022).

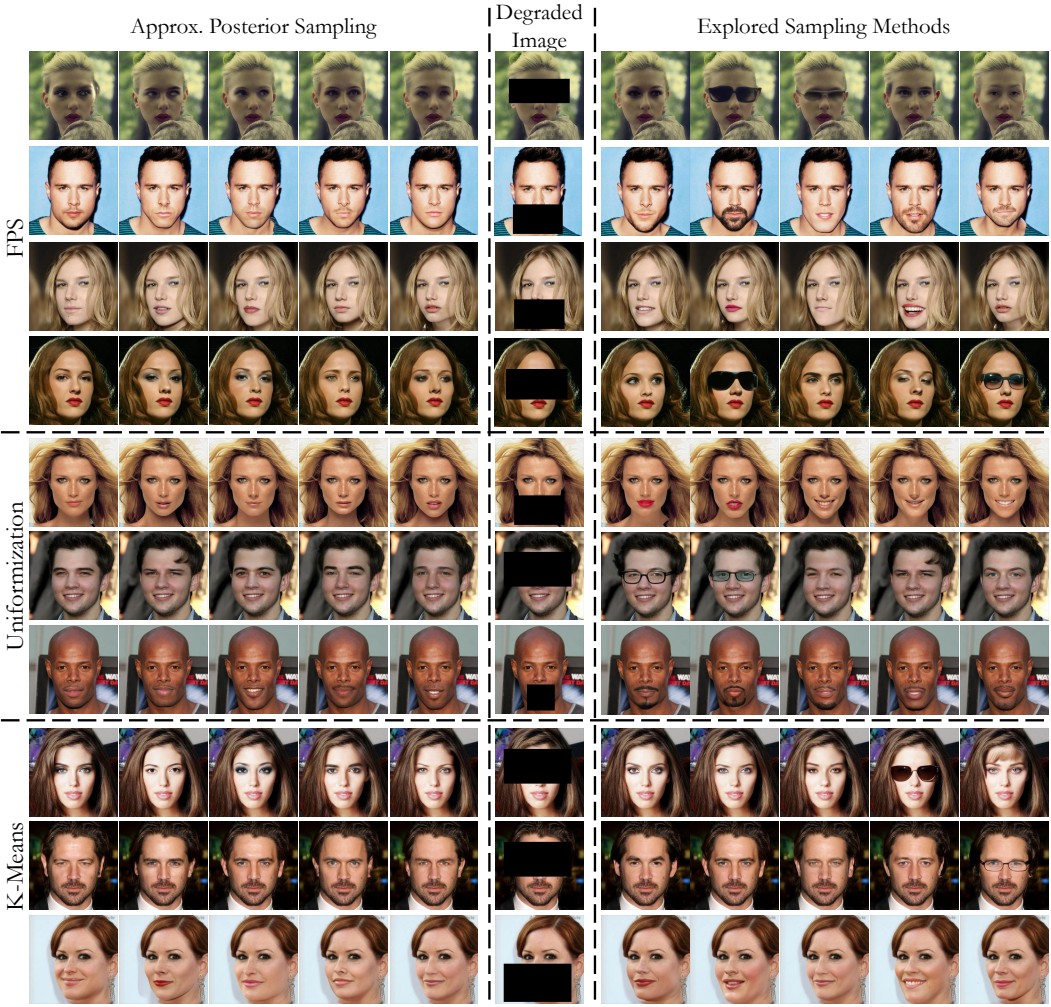

Figure 30: **Additional comparisons of inpainting on CelebAMask-HQ with sub-sampling approaches vs. using the approximate posterior**. Restorations created using RePaint Lugmayr et al. (2022a).

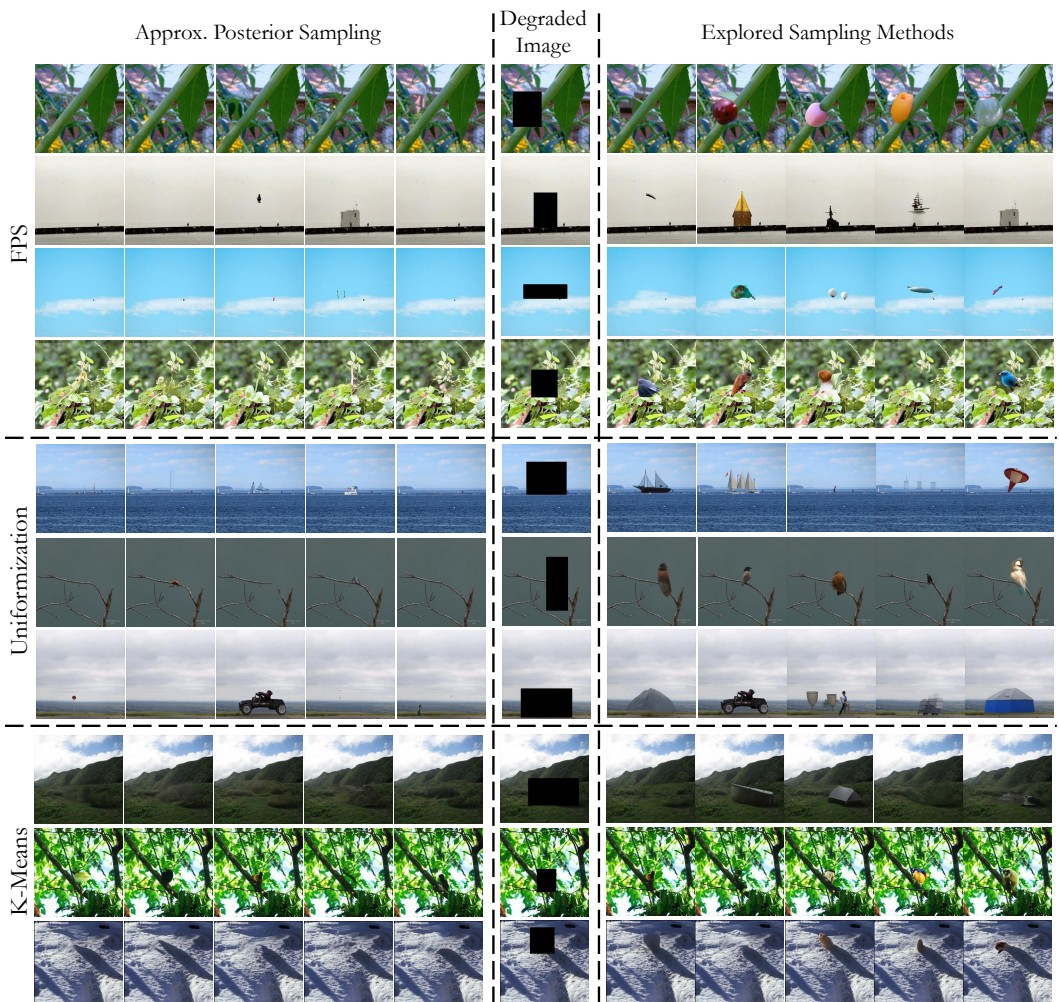

Figure 31: **Additional comparisons of inpainting on PartImagenet with sub-sampling approaches vs. using the approximate posterior**. Restorations created using RePaint Lugmayr et al. (2022a).

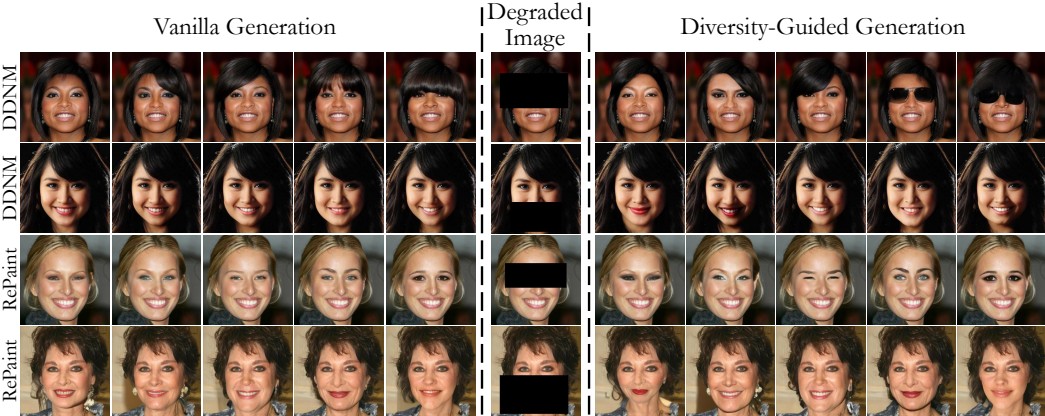

Figure 32: **Additional comparisons of inpainting on CelebAMask-HQ, with and without diversity-guidance**. Restoration method marked on the left.

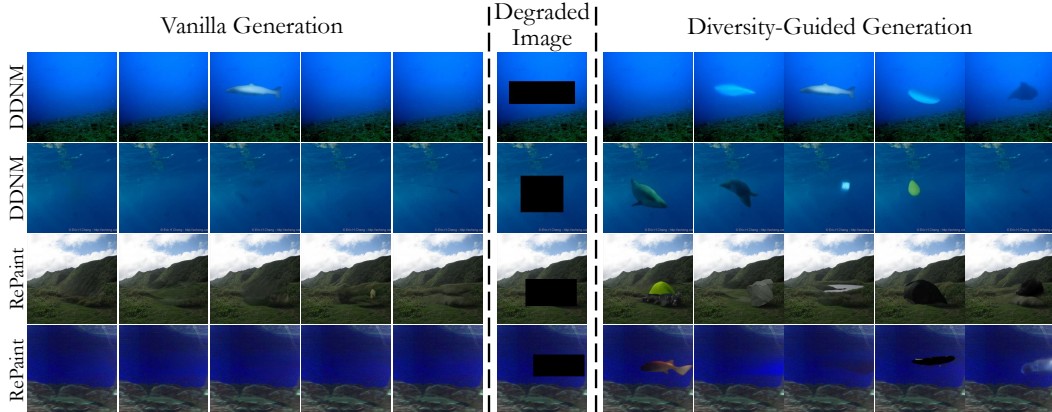

Figure 33: **Additional comparisons of inpainting on PartImageNet, with and without diversity-guidance**. Restoration method marked on the left.

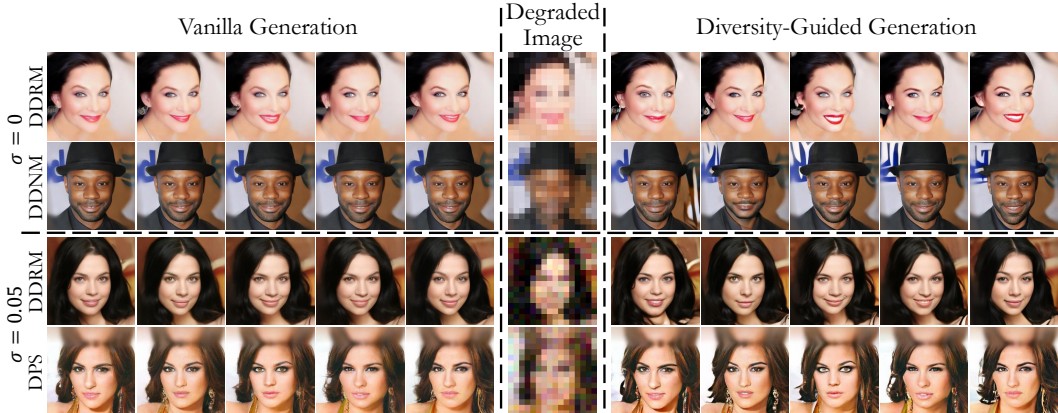

Figure 34: **Additional comparisons of noisy and noiseless super-resolution on CelebAMask-HQ, with and without diversity-guidance**. Restoration method and noise level marked on the left.

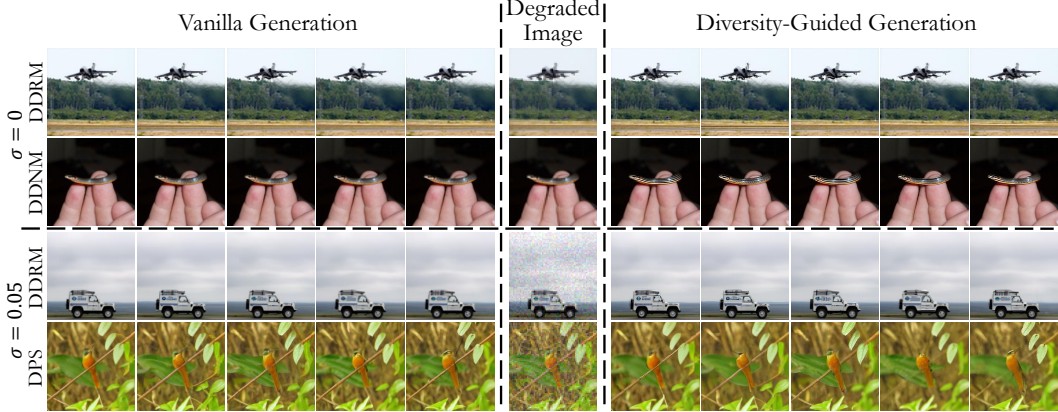

Figure 35: **Additional comparisons of noisy and noiseless $4\times$ super-resolution on PartIma-geNet, with and without diversity-guidance**. Restoration method and noise level marked on the left.

