# OpenReview forum: "From Posterior Sampling to Meaningful Diversity in Image Restoration"
_ICLR.cc/2024/Conference — ICLR 2024 poster_

### Official Review · Reviewer_Fph6 · 2023-10-26

**Soundness:** 3 good
**Presentation:** 3 good
**Contribution:** 3 good
**Rating:** 8
**Confidence:** 3

**Summary:**

The current trend in image restoration methods involves sampling multiple outputs from the posterior distribution rather than generating a single one. This paper has highlighted a fundamental limitation of posterior sampling, specifically in generating semantically distinct possibilities using a reasonably small number of samples. This limitation arises from the heavy-tailed nature of the posterior distribution along semantically relevant directions. As a solution, the paper suggests creating compositions of small but meaningfully diverse outputs. The study also delves into a comprehensive examination of what constitutes a set of reconstructions that are meaningfully diverse. To achieve this, the paper explores various post-processing approaches, including Uniformization, K-Means Centers, and Farthest Point Strategy, to obtain semantically meaningful and diverse outputs using existing image restoration methods. Furthermore, the paper introduces a practical approach for enabling diffusion-based image restoration methods to produce outputs that are meaningfully diverse. The effectiveness of these methods is validated through quantitative measures and user studies, demonstrating their superiority over conventional posterior sampling techniques.

**Strengths:**

-  identify an inherent conceptual issue with posterior sampling and elucidate the root cause of this problem
- conduct an analysis to determine the criteria for meaningful diversity. This analysis involves an exploration of three fundamental strategies for diverse sub-sampling.
-  introduce a novel diffusion-based generation strategy, which offers a practical means to achieve restoration results that are meaningfully diverse.

**Weaknesses:**

None

**Questions:**

I have some concerns while reviewing the main paper, but they have been addressed in the supplementary material.

A suggestion for the authors: The discussion on future work could be expanded, e.g., what specific desired variations can be for potential research in the field? This would provide more clarity and guidance for researchers interested in pursuing further work in this area.

---

> ### Author Response · Authors · 2023-11-19
> **Response to reviewer Fph6**
>
> Thank you for your positive feedback. We believe there exist several interesting directions for future research. In particular, in this paper we investigate general meaningful diversity, which focuses on exploring different kinds of diversity at once. For example, in the context of restoration of face images, we aim for our representative set to cover diverse face structures, glasses, makeup, etc.
> However, for certain applications it can be desirable to reflect the diversity for a specific property, e.g. covering multiple types of facial hair and accessories while keeping the identity fixed, or covering multiple identities while keeping the facial expression fixed.
> The ability to achieve diversity in only specific attributes can potentially be important in e.g. the medical domain, for example to allow a radiologist to view a range of plausible pathological interpretations for a specific tumor in a CT scan, or to present a forensic investigator with a representative subset of headwear that are consistent with a low quality surveillance camera footage. \
> We included this in the revised manuscript.

---

> > ### Comment · Reviewer_Fph6 · 2023-11-22
> >
> > Thank you for your response. I will keep the rating.

---

### Official Review · Reviewer_iJud · 2023-10-29

**Soundness:** 3 good
**Presentation:** 3 good
**Contribution:** 2 fair
**Rating:** 6
**Confidence:** 5

**Summary:**

This paper is concerned with  generating  diverse solutions to ill-posed image restoration problems. Usual random sampling from the heavy tailed posterior distribution will have samples from high density regions with a much higher probability than from low density regions, making it an impractical approach to generate diverse solutions from limited samples. Given a large number of samples from posterior, the paper explores three approaches to extract limited number of samples representing meaningful diversity. Further, the paper proposes a simple guidance mechanism to improve diversity of solutions. This involves running the diffusion process to simultaneously generate multiple images starting from different noise samples, and encouraging dissimilarity between estimates of clean images at every step. This guidance improves diversity of solutions of recent diffusion based restoration methods while  maintaining reconstruction quality.

**Strengths:**

The paper is well motivated and well-written.

The proposed technique is simple and straightforward to implement and the work is reproducible.

The proposed method improves the diversity of solutions.

**Weaknesses:**

The observation that random sampling leads to limited diversity for long tailed distributions was also noted in [a] in the context of image generation.  [a] also proposes to modify the sampling process in diffusion models to generate samples from low density regions of a long tailed distribution. Though the focus in [a] is on generation, and not restoration, it is still a relevant work, and could be cited by the authors.

The authors could also cite {b] which attempts to generate diverse solutions to linear inverse problems using pretrained gans.

While the proposed guidance method improves diversity and is useful, the technical contribution  seems rather limited. This is the reason for my rating.

[a] Sehwag et al. "Generating high fidelity data from low-density regions using diffusion models." In CVPR 2022

[b]  Montanaro etal. "Exploring the solution space of linear inverse problems with gan latent geometry." In  IEEE International Conference on Image Processing  (ICIP) 2022

**Questions:**

Could the authors clarify the following issues:

It is not clarified what kind of down-sampling is used to obtain degraded images for super-resolution.

Fig. 14, degraded input for super-resolution looks wierd, the results of DPS look like down-sampled images.

Figs 28 , 29 super-resolution or inpainting. degraded image looks like a low resolution image, caption says inpainting.

---

> ### Author Response · Authors · 2023-11-19
> **Response to reviewer iJud**
>
> Thank you for the constructive review.
>
> 1. **Relevant citations:**\
> Thanks for pointing us to these relevant works. In the updated manuscript, we added the papers by Sehwag et al. and by Montanaro et al. into a related work paragraph dedicated to works that enhance perceptual coverage.
>
> 2. **Our technical contribution:**\
> Please note that the contribution of our paper is broader than the proposed guidance method itself: We are the first to systematically explore what it means for a set of reconstructions to be meaningfully diverse by proposing a thorough analysis backed by human user studies. Moreover, we illustrate the heavy-tailed behavior of the posterior quantitatively, and frame this as an inherent limitation of posterior sampling in achieving meaningful diversity. Of course, we eventually also accompany the observations with a practical method for obtaining a small representative set of solutions to any inverse problem. Our method uses the well established tool of diffusion guidance. But the novelty lies mainly in the context in which we use this tool.
>
> 3. **Additional questions/issues:**\
> Thanks for pointing out these issues. We address them below and have revised the manuscript to clarify them:
>
>    a. *Downsampling kernel*: In all super-resolution experiments, we use a bicubic downsampling kernel to obtain the degraded LR image, and add white Gaussian noise to the low-resolution image (with the mentioned noise level) for the noisy SR experiments. We agree that this was not clearly stated. We added this information to Sec. 4.1 and also to App. C.
>
>    b. *Fig. 14 in the supplementary*: We regret the confusion. We have now revised the figure to show, for each example, the degraded image on the left, and its corresponding model output on the right. We updated the caption accordingly.
>
>    c. *Figures 28 & 29 (now 34,35)*: Thanks. We fixed the captions to reflect the correct figure contents.

---

> > ### Comment · Reviewer_iJud · 2023-11-21
> > **Response and final rating.**
> >
> > Thank you for your response. I have now updated my rating.

---

### Official Review · Reviewer_ppuX · 2023-10-30

**Soundness:** 4 excellent
**Presentation:** 3 good
**Contribution:** 4 excellent
**Rating:** 8
**Confidence:** 4

**Summary:**

This paper discusses the problem of generating a diverse set of reconstructed samples when solving an imaging inverse problem using a pretrained diffusion model. The traditional setup is to cast the inverse problem as one of posterior distribution sampling. However, as the authors discussed this may lead to a concentrated set of samples where most of them look very similar to each other. The paper discusses different ways of measuring and modeling diversity in image reconstruction and later introduces a method for generating a given number of samples  conditioned on a low-resolution / low-quality image, maximizing diversity and using a pretrained diffusion model. The method is evaluated on popular benchmarks on two restoration tasks: super-resolution (4/16x, noisy and noiseless) and inpainting.

**Strengths:**

* The paper is generally well-written, address an interesting and relevant problem with a clear narrative and presentation.
* The technical contribution of the paper is good: analizes a problem and propose a practical solution. This could lead to more research on this problem ("meaningfully diverse image restoration")

**Weaknesses:**

Not weakness per se, more like observations:
* Experimental results are interesting but limited. For example, image colorization is one of the main problems where diversity reconstruction could have a significant impact.
* Presentation could be a little more balanced. In particular, the experimental section (Sec 6) could be a little longer and maybe show more results, discuss a little more some of the algorithmic decisions (presented in Sec 5). The authors opted to put more content on Sec 3 and 4, which seems also a valid decision.

**Questions:**

The paper is in general well-written, addresses an interesting problem, and present an interesting solution. So I'm in favour of accepting this paper as it is now. However, if the authors would like to make the paper better I list a few questions/comments that could be helpful.

1.  **Analysis of the proposed method**. The method seems to do the job, but I wonder if the authors could provide more insight on their proposed solution. For example, a few alternative designs could be:
    * Instead of increasing distance with respect to NN, just maximize the sample variance (or average pairwise distance as in the reported LPIPS metric)
    * What happens with the sample average. In the traditional formulation, the average of the generated samples should be close to the posterior mean (which is also another possible estimator that minimizes distortion). Have you tried to compute and compare the sample means w/wo the diversified guidance ? I guess that the mean wouldn't be as good as in the traditional case, but maybe a more robust mean estimator could lead to a very similar result (an estimator that minimizes distortion)

2. **Other Experiments**. In particular regarding Colorization (e.g., one very relevant practical problem is face colorization, can we generate a meaningful diverse set of reconstructed images when starting from a gray photo of a face?)

3. **Connections to other work on GANs** (and mode collapse which seems related).

    * Mao, Q., Lee, H.Y., Tseng, H.Y., Ma, S. and Yang, M.H., 2019. Mode seeking generative adversarial networks for diverse image synthesis. In Proceedings of the IEEE/CVF conference on computer vision and pattern recognition (pp. 1429-1437).

   * Yu, N., Li, K., Zhou, P., Malik, J., Davis, L. and Fritz, M., 2020. Inclusive gan: Improving data and minority coverage in generative models. In Computer Vision–ECCV 2020: 16th European Conference, Glasgow, UK, August 23–28, 2020, Proceedings, Part XXII 16 (pp. 377-393). Springer International Publishing.

4. **Training**. This paper is not about training models that allow to generate a diverse set of reconstructions. But, I wonder if the authors could give a comment on wether we could incorporate the idea of diverse sample generation also during training. In particular I wonder if the authors could comment on wether this is an interesting thing to consider, for example in the context of conditional generation ([SR3](https://iterative-refinement.github.io/), [Palette](https://iterative-refinement.github.io/palette/)) or current Bridge restoration models ([InDI](https://openreview.net/forum?id=VmyFF5lL3F), [I2SB](https://i2sb.github.io/))

---

> ### Author Response · Authors · 2023-11-19
> **Response to reviewer ppuX**
>
> We thank the reviewer for the insightful comments and questions.
> 1. **Analysis of the proposed method:**
>
>    a. *Alternatives to increasing distance w.r.t. Nearest Neighbor*: Thanks for this good point. We actually experimented with several alternatives for increasing diversity, but found them all to lead to similar results. This is the reason we eventually decided to go with the simple nearest-neighbor (NN) solution. In the updated manuscript, we added to App. F a few comparisons between NN guidance and guidance that pushes the samples away from their average. As can be seen, both lead to good results, and the differences between them are small.
>
>    b. *The effect on the sample average*: Following your question, we compared the RMSE of the average of our diversity-guided samples to the RMSE of the average of the vanilla samples. We found that the former is only 0%-2% larger than the latter (e.g. an increase of 0.63% in RMSE for SR with DDRM on the CelebA-HQ dataset). This relates to the previous point you raised, as it implies that although our samples are more spread than the vanilla samples, they are still spread roughly around the same point as the vanilla sample  – the posterior mean.
>
> 2. **Other experiments:**\
> Thanks. Following your suggestion, we applied our diversity guided process (using DDNM) to image colorization for the CelebA-HQ and Imagenet datasets. We added example results to App. I.1 in the updated manuscript, where we indeed see a significant diversity gain with our method compared to the vanilla reconstruction.
>
> 3. **Connections to other work on GANs:**\
> These GAN papers are indeed relevant. We added a discussion in the related work section about these and other related works, under a paragraph titled “Enhancing perceptual coverage”.
>
> 4. **Training:**\
> Incorporating a diversity encouraging loss within a (conditional) diffusion model is actually a very interesting theoretical research avenue. We do not currently have a concrete direction for how this should be done. But we should note that achieving the diversification effect using guidance, as we do here, has certain advantages over training. In particular, it allows the same diffusion model to be used at test time either with or without guidance. Although the latter is not optimal for communicating uncertainty (as we show), it may be needed in certain specific applications, e.g. for unbiased quantitative analysis of the posterior, like computing confidence intervals along the principal components of the posterior (as done in [R1]).
>
> [R1] Belhasin, Romano, Freedman, Rivlin, Elad, “Principal Uncertainty Quantification with Spatial Correlation for Image Restoration Problems”, arXiv:2305.10124, 2023

---

> > ### Comment · Reviewer_ppuX · 2023-11-22
> > **Re: Response to reviewer ppuX**
> >
> > Thank you for your response. I think this is a good paper so I'm keeping my score.

---

### Meta-Review · Area_Chair_3zJd · 2023-12-13

**Metareview:**

This paper aims to address the limitations of sample distribution concentration caused by posterior distribution sampling, proposing three methods to extract samples representing meaningful diversity. Additionally, it introduces a simple guide to encourage diversity generation, initializing different noises, and ensuring the generated images do not converge. The methodology is easy to understand, and the experimental results are clearly demonstrated. However, the paper has some shortcomings in motivation and method contribution.

**Justification For Why Not Higher Score:**

The motivation for diversity in image restoration is to accommodate the pathological issue of low-quality images corresponding to multiple source images, which seems somewhat tenuous in connection with semantic diversity generation. Moreover, there is an overlap in motivation with the previous work "Generating high fidelity data from low-density regions using diffusion models," thus limiting the paper's contribution in this aspect. Additionally, while the proposed method is effective in generating diversity, the technical contribution of the paper is limited.

**Justification For Why Not Lower Score:**

All reviewers provided positive or slightly positive comments. The experimental results of the paper are very interesting, and the technical explanation is easy to understand and straightforward to implement.

---

### Decision · Program_Chairs · 2024-01-16

Accept (poster)